# IMPUTATION FOR PREDICTION: BEWARE OF DIMINISHING RETURNS.

**Marine Le Morvan**
Soda, Inria Saclay
`marine.le-morvan@inria.fr`

**Gaël Varoquaux**
Soda, Inria Saclay
`gael.varoquaux@inria.fr`

## ABSTRACT

Missing values are prevalent across various fields, posing challenges for training and deploying predictive models. In this context, imputation is a common practice, driven by the hope that accurate imputations will enhance predictions. However, recent theoretical and empirical studies indicate that simple constant imputation can be consistent and competitive. This empirical study aims at clarifying *if* and *when* investing in advanced imputation methods yields significantly better predictions. Relating imputation and predictive accuracies across combinations of imputation and predictive models on 19 datasets, we show that imputation accuracy matters less i) when using expressive models, ii) when incorporating missingness indicators as complementary inputs, iii) matters much more for generated linear outcomes than for real-data outcomes. Interestingly, we also show that the use of the missingness indicator is beneficial to the prediction performance, *even in MCAR scenarios*. Overall, on real-data with powerful models, improving imputation only has a minor effect on prediction performance. Thus, investing in better imputations for improved predictions often offers limited benefits.

## 1 INTRODUCTION

Databases are often riddled with missing values due to faulty measurements, unanswered questionnaire items or unreported data. This is typical of large health databases such as the UK Biobank (Sudlow et al., 2015), the National Health Interview Survey (Blewett et al., 2019) and others (Perez-Lebel et al., 2022). Statistical analysis with missing values has been widely studied, particularly to estimate parameters such as means and variances (Little & Rubin, 2019). However, how to best deal with missing values for prediction has been less studied. Since most machine learning models do not natively handle missing values, common practice is to impute missing values before training a model on the completed data, often with the expectation that "good" imputation improves predictions. Considerable efforts have been dedicated to improving imputation techniques, utilizing Generative Adversarial Networks (Yoon et al., 2018), Variational AutoEncoders (Mattei & Frellsen, 2019), optimal transport (Muzellec et al., 2020) or AutoML-enhanced iterative conditional imputation (Jarrett et al., 2022) among others. Most of these studies concentrate on imputation accuracy without assessing performance on subsequent tasks. However, theoretical arguments suggest that good imputation is not needed for good prediction (Le Morvan et al., 2021; Josse et al., 2019). These arguments are asymptotic and whether they hold in typical cases is debatable. To address the discrepancy between this theory and the emphasis on imputation efforts, there is a critical need for empirical studies to determine whether better imputations actually lead to better predictions.

Theory does establish that in some scenarios, better imputations imply better predictions. For instance with a linearly-generated outcome, the optimal prediction is a linear model of the optimally-imputed data (Le Morvan et al., 2021). Thus, when using a linear model for prediction, better imputations generally lead to better predictions. However, theoretical results suggest that in very high-dimensional settings (Ayme et al., 2023) or in small dimensions with uncorrelated features (Ayme et al., 2024), simple constant imputations can be sufficient for linear models. Beyond linear models, empirical studies on real data have shown the competitiveness of simple imputations –such as the mean– (Paterakis et al., 2024; Perez-Lebel et al., 2022; Shadbahr et al., 2023; Luengo et al., 2012) aligning with theoretical arguments. However, their findings may be driven by "predictive" missingness

(Missing Not At Random data Little & Rubin, 2019; Josse et al., 2019), for which most imputation methods are invalid.

Drawing robust and broadly applicable conclusions from existing empirical research is challenging, as several key experimental factors can influence conclusions. One critical factor is the missingness mechanism, which falls into three categories: MCAR (Missing Completely At Random), MAR (Missing At Random), and MNAR (Missing Not At Random). MCAR is the simplest case, where missing entries occur with a fixed probability, independent of observed or unobserved data. In MAR, missingness depends solely on observed variables, whereas in MNAR, it is related to the unobserved values themselves, making it informative. Naturally occurring missing values are generally assumed to be MNAR. However, most imputation algorithms are not valid in MNAR scenarios, questioning the utility of advanced imputation algorithms in such cases. MCAR allows for higher quality imputations but their benefit for downstream performance needs to be evaluated. Beyond the missingness mechanism, other influential factors include (i) the missing rate - low proportions of missing values may have a limited impact on predictive performance; (ii) the use of missingness indicators - concatenating a binary indicator for missing values to the original sample is a common practice for prediction, as implemented in scikit-learn (Pedregosa et al., 2011); (iii) the choice of downstream model - more flexible models may better compensate for poor imputation; and (iv) the proportion of categorical features - for categorical features, the most effective approach is often to treat missing values as a separate category, bypassing imputation altogether.

To draw actionable conclusions taking into account these potentially influential factors, we *quantify the change in predictive performance resulting from a gain in imputation accuracy across controlled experimental settings*. First, we focus on MCAR as a best-case scenario since it allows for high-quality imputations that are most likely to improve downstream performance. If only limited gains are observed under these ideal conditions, the benefits in general scenarios are likely smaller. Thus we aim to establish an upper bound on the potential benefits of imputation. Results under MNAR conditions are nonetheless included (subsection 4.4 and Appendix L) and confirm that imputation provides less benefits in such cases. Following a similar rationale, we focus on numerical features, where imputation quality is most likely to affect downstream performances, as missing categorical values are better handled as a separate category. As for the remaining factors, we vary the choice of downstream prediction model, the use of the missingness indicator, as well as the missing rate to condition our conclusions on the exprimental setting. Rather than identifying the "best" imputation and prediction pipeline, our goal is to rigorously assess the impact of imputation quality on predictive performance.

Section 2 introduces related work, covering both benchmarks for prediction with missing values and available theory. Section 3 details our experimental procedures, specifically the methods examined. Section 4 presents our findings, relating gains in imputation to prediction performance. Finally, section 5 summarizes the lessons learned.

## 2    RELATED WORK

**Benchmarks.**    Several benchmark studies have investigated imputation in a prediction context (Paterakis et al., 2024; Jäger et al., 2021; Ramosaj et al., 2022; Woźnica & Biecek, 2020; Perez-Lebel et al., 2022; Poulos & Valle, 2018; Shadbahr et al., 2023; Li et al., 2024; Luengo et al., 2012; Bertsimas et al., 2024). However, drawing definitive conclusions from most studies is challenging due to various limitations in scope and experimental choices. For example, Bertsimas et al. (2018); Li et al. (2024) trained imputation methods using both the training and test sets, rather than applying the imputation learned on the training set to the test set, which is not possible with many imputation packages. This approach creates data leakage. Woźnica & Biecek (2020) trained imputers separately on the train and test sets, which creates an "imputation shift" —a situation where the imputation patterns between the train and test sets differ, causing inconsistencies in the data used for model training versus model evaluation. Jäger et al. (2021) discards and imputes values in a single column of the test set, chosen at random and fixed throughout the experiments. Yet as they note, conclusions can change drastically depending on the importance of the to-be-imputed column for the prediction task or its correlation with other features. Some studies (Poulos & Valle, 2018; Ramosaj et al., 2022) use a small number of datasets (resp. 2 and 5 datasets from the UCI machine learning repository respectively), thus limiting the significance of their conclusions. Woźnica & Biecek (2020) do

not perform hyperparameter tuning for the prediction models, while Ramosaj et al. (2022) tunes hyperparameters on the complete data, though it is unclear whether the best hyperparameters on complete data are also the best on incomplete data. Furthermore, some benchmarks focus on specific types of downstream prediction models, such as linear models (Jäger et al., 2021), AutoML models (Paterakis et al., 2024) or Support Vector Machines (Li et al., 2024), meaning their conclusions should not be generalized to all types of downstream models. Finally, only Perez-Lebel et al. (2022) and Paterakis et al. (2024) evaluate the use of the missingness indicator as complementary input features.

Among the benchmarks with largest scope, Paterakis et al. (2024) recommend mean/mode imputation with the indicator as the default option in AutoML settings, both for native and simulated missingness. It is among the top-performing approaches, never statistically significantly outperformed, and is also the most cost-effective. They also show that using the missingness indicator as input improves performances slightly but significantly for most imputation methods on naturally-occurring missingness. Perez-Lebel et al. (2022) focus on predictive modeling for large health databases, which contain many missing values. They compare various imputation strategies (mean, median, k-nearest neighbors imputation, MICE) combined with gradient-boosted trees (GBTs) for prediction, as well as GBTs with native handling of missing values. Similarly to Paterakis et al. (2024), they find that appending the indicator to the imputed data significantly improves performances, which may reflect MNAR data. While they recommend resorting to the native handling of missing values as it is relatively cheap, their results further indicate that no method is significantly better than using the mean as imputation method together with the indicator. Shadbahr et al. (2023) also find that the best imputations do not necessarily result in the best downstream performances. Using an analysis of variance, they show that the choice of imputation method has a significant but small effect on the classification performance.

Whether better imputation leads to better prediction may vary depending on factors like the choice of downstream model, the missingness rate, or the characteristics of the datasets. Yet, many studies seek a definitive conclusion across diverse settings. Only Paterakis et al. (2024) conducted a meta-analysis, but it did not determine when more advanced imputation strategies are beneficial compared to mean or mode imputations. Identifying scenarios in which better imputations are more likely to improve predictions is however of strong practical interest.

**Theoretical insights.** Previous works have addressed this question from a theoretical point of view. Le Morvan et al. (2021) showed that for all missingness mechanisms and almost all deterministic imputation functions, universally consistent algorithms trained on imputed data asymptotically achieve optimal performances in prediction. This is in particular true for simple imputations such as the mean (Josse et al., 2019), thereby providing rationale to favor simple imputations over more accurate ones. Essentially, optimal prediction models can be built on mean-imputed data by modeling the mean as a special value encoding for missingness. Ayme et al. (2023) also provide theoretical support for the use of simple imputations, as they advocate for the use of zero imputation in high-dimensional settings. They show that, for a linear regression problem and MCAR missingness, learning on zero-imputed data instead of complete data incurs an imputation bias that goes to zero when the dimension increases. This holds given certain assumptions on the covariance matrix, which intuitively impose some redundancy among variables. Finally, Van Ness et al. (2023) prove that in MCAR settings, the best linear predictor assigns zero weights to the missingness indicator, whereas these weights are non-zero in MNAR settings. Their theoretical results imply that the missingness indicator neither degrades nor enhances performances asymptotically in MCAR.

## 3 EXPERIMENTAL SETUP.

**Imputation methods.** We chose four imputation models to *cover a wide range of imputation qualities*, in order to facilitate the estimation of effects and correlations between imputation and prediction accuracies.

**mean** - each missing value is imputed with the mean of the observed values in a given variable. It provides a useful baseline for assessing the effectiveness of advanced techniques.

**iterativeBR** - each feature is imputed based on the other features in a round-robin fashion using a Bayesian ridge regressor. This method is related to `mice` (Van Buuren & Groothuis-Oudshoorn, 2011) as it also relies on a fully conditional specification (Van Buuren, 2018). It is implemented in `scikit-learn`'s `IterativeImputer` (Pedregosa et al., 2011).

**missforest** (Stekhoven & Bühlmann, 2012) - operates in a manner analogous to `iterativeBR`, wherein it imputes one feature using all others and iteratively enhances the imputation by sequentially addressing each feature multiple times. The key distinction lies in its utilization of a random forest for imputation rather than a linear model. We used `scikit-learn`'s `IterativeImputer` with `RandomForestRegressor` as estimators. Default parameters for Missforest were set to `n_estimators=30` and `max_depth=15` for the random forests (the higher the better) to keep a reasonable computational budget. Note that in `HyperImpute` (Jarrett et al., 2022), random forests are more limited: 10 trees and a maximum depth of 4.

**condexp** - uses the conditional expectation formula of a multivariate normal distribution to impute the missing entries given the observed ones. The mean and covariance matrix of the multivariate normal distribution are estimated with (pairwise) available-case estimates (Little & Rubin, 2019, section 3.4), i.e., the $(i, j)^{th}$ entry of the covariance matrix is estimated solely from samples where both variables $i$ and $j$ are observed. This approach offers computational advantages over more resource-intensive approaches such as the Expectation-Maximization (EM) algorithm. It is related to Buck's method (Buck, 1960; Little & Rubin, 2019, section 4.2).

Mean imputation can be expected to give the worst imputation, with other methods offering varying improvements. In particular, `missforest` often delivers top-tier performance on tabular data (Waljee et al., 2013; Jarrett et al., 2022; Yoon et al., 2018; Mattei & Frellsen, 2019; Jäger et al., 2021).

**Models:** As the effect of imputation on prediction quality can be modulated by the predictive model used, we included three predictive models. We took care to include both a deep learning and a tree-based representative, as the prediction functions produced by these models have different properties, for example regarding their smoothness. These representatives were chosen because they were identified as state-of-the-art in their category according to recent benchmarks (Borisov et al., 2022; Grinsztajn et al., 2022).

- **MLP**: a basic Multilayer Perceptron with ReLU activations, to serve as a simple baseline.
- **SAINT** (Somepalli et al., 2021): Self-Attention and Intersample Attention Transformer (SAINT) is a deep tabular model that performs both row and column attention. The numerical features are first embedded to a $d$-dimensional space before being fed to the transformer. We chose SAINT as it has been shown to be state-of-the-art among deep learning approaches for tabular data in several surveys (Borisov et al., 2022; Grinsztajn et al., 2022).
- **XGBoost** (Chen & Guestrin, 2016): We chose XGBoost as it is a popular state-of-the-art boosting method, and it has been shown to be the best tree-based model on regression tasks with numerical features only in Grinsztajn et al. (2022).

For XGBoost and the MLP, hyperparameters were tuned using Optuna (Akiba et al., 2019) with 50 trials, i.e, Optuna draws 50 sets of hyperparameters, trains a model for each of these hyperparameter sets, and retains the best one according to the prediction performance on the validation set. For SAINT, we used the default hyperparameters provided by its authors (Somepalli et al., 2021) for computational reasons. Tables 2 to 4 in the Appendix provide the hyperparameter spaces searched, default hyperparameters as well as optimization details.

**Native handling of missing values:** Both SAINT and XGBoost can directly be applied on incomplete data, without prior imputation of the missing values, each with its own strategy. XGBoost uses the Missing Incorporated in Attribute (MIA) (Twala et al., 2008; Josse et al., 2019) approach. When splitting, samples with missing values in the split feature can go left, right, or form their own leaf. MIA retains the option that minimizes the prediction error. In SAINT, numerical features are embedded in a $d$-dimensional space using simple MLPs. In case of missing value, a learnable $d$-dimensional embedding is used to represent the `NaN`. Each feature has its own missingness embedding.

**The datasets** We use a benchmark created by Grinsztajn et al. (2022) for tabular learning. It comprises 19 datasets (listed in table 1), each corresponding to a regression task with continuous and ordinal features. Missing data is generated according to a MCAR mechanism with either 20% or 50% missing rate. Specifically, each value has a 20% or 50% chance of being missing. Continuous features are gaussianized using scikit-learn's `QuantileTransformer` while ordinal features are standard scaled to have a zero mean and unit variance. This is true for all imputation and model combination except for XGBoost with native handling of missing values, as it is not expected to benefit from a normalization. The outputs $y$ are also standard scaled. In all cases, the parameters of these data

normalizations are learned on the train set with missing values. We also provide experiments on semi-synthetic data where the response $y$ is simulated as a linear function of the original data $X$. The coefficients $\beta$ of the linear function are all taken equal and scaled so that the variance of $\beta^\top X$ is equal to 1. Noise is added with a signal-to-noise ratio of 10.

**Evaluation strategy**   Each dataset is randomly split into 3 folds (train - 80%, validation - 10% and test - 10%), and each split is furthermore capped at 50,000 samples (table 1). Train, validation and test sets are imputed using the same imputation model trained on the train set. Prediction models are then trained on the imputed train set. When the indicator is used, it is appended as extra features to the imputed data, it is not leveraged for the imputation stage. We run all combinations of the 3 prediction models with the 4 imputation techniques, with and without the indicator, resulting in $4 \times 3 \times 2 = 24$ models to which we add XGBoost and SAINT with native handling of missing values. This results in a total of 26 models displayed in Figure 1. Finally, the whole process is repeated with 10 different train/validation/test splits. For reproducibility, the code is available at `https://github.com/marineLM/Imputation_for_prediction_benchmark`.

**Computational resources.**   Properly benchmarking methodologies with missing values is particularly resource-intensive, as already emphasized in previous works (Jäger et al., 2021; Perez-Lebel et al., 2022). Computing costs are driven on the one hand by the need to run the imputation and prediction pipeline across multiple train-test splits (which is important to account for benchmark variance, Bouthillier et al., 2021), and on the other hand by the combinatorics of imputation and prediction models, hyperparameter optimization for both, inclusion and exclusion of the missingness indicator, and varying missing rates. Multiplying the number of models (26) with the number of datasets (19 + 19 linear versions), the hyperparameter tuning (50 trials), the number of repetitions of the experiments (10), and the 2 missing rates, we get a very large number of runs (around 1,000,000). As some methods are computationally expensive –such as missforest for imputing, as well as SAINT notably when the indicator is used–, these experiments required a total of 325 CPU days for the MCAR experiments. A fifth of this time was dedicated to imputation.

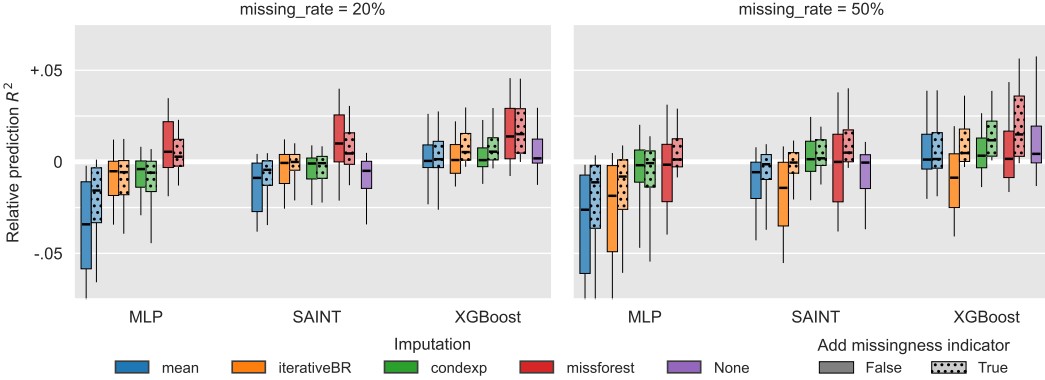

Figure 1: **Relative prediction performances across datasets for different imputations, predictors, and use of the missingness indicator.** Each boxplot represents 200 points (20 datasets with 10 repetitions per dataset). The performances shown are $R^2$ scores on the test set relative to the mean performance across all models for a given dataset and repetition. A value of 0.01 indicates that a given method outperforms the average performance on a given dataset by 0.01 on the $R^2$ score. Corresponding critical difference plots in figs. 6 and 7.

## 4 RESULTS: DETERMINANTS OF PREDICTIONS PERFORMANCE WITH MCAR MISSINGNESS

### 4.1 BENEFITS OF SOPHISTICATED IMPUTATION, THE INDICATOR, AND XGBOOST AMID HIGH VARIANCE.

Figure 1 summarizes the relative performance of the various predictors combined with the different imputation schemes across the 19 datasets. Some trends emerge: more sophisticated imputers tend to improve prediction, with missForest-based predictors often outperforming those using condexp or iterativeBR imputers, which in turn outperform predictors based on mean imputation. However, using the missingness indicator decreases this effect. Additionally, while less powerful models like MLPs show greater improvements with more advanced imputations, this effect is barely noticeable for the strongest predictor, XGBoost, which maintains its advantages on tabular data (as described in Grinsztajn et al., 2022) even in the presence of missing values.

That the best predictor barely benefits from fancy imputers brings us back to our original question: should efforts go into imputation? Drawing a conclusion from figure 1 would be premature: the variance across datasets is typically greater than the difference in performance between methods (critical difference diagram in figs. 6 and 7). For example, missforest + XGBoost + indicator outperforms all other methods in only 4 out of 19 datasets. Additionally, XGBoost + indicator does not perform significantly better with missforest than with condexp at 50% missingness, while mean imputation does not always lead to the worst prediction. In what follows, we focus on quantifying the effects of improved imputation accuracy on predictions in different scenarios.

### 4.2 A DETOUR THROUGH IMPUTATION ACCURACIES: HOW DO IMPUTERS COMPARE?

Although comparing imputers is not our main objective, it is enlightening for our prediction purpose to characterize their relative performance range. Figure 2 (left) gives imputation performances measured as the $R^2$ score between the imputed and ground truth values relative to the average across imputations, for each dataset. At a 20% missing rate, missForest is the best imputer, followed by condexp and iterativeBR, which are nearly tied, while mean imputation performs significantly worse. At a 50% missing rate, the imputation accuracy of all but mean imputation drop, but condexp is much less affected. It is interesting that such a simple method performs best. It is notably two orders of magnitude faster than missforest (figure 2 right), which makes it an imputation technique worth considering. It is possible that the gaussianization of the features helped condexp, although a feature-wise gaussianization does not produce a jointly Gaussian dataset.

In order to highlight the link between imputation and prediction quality, it is necessary to achieve varying imputation qualities. Here the high range of imputation accuracy between the best and worst

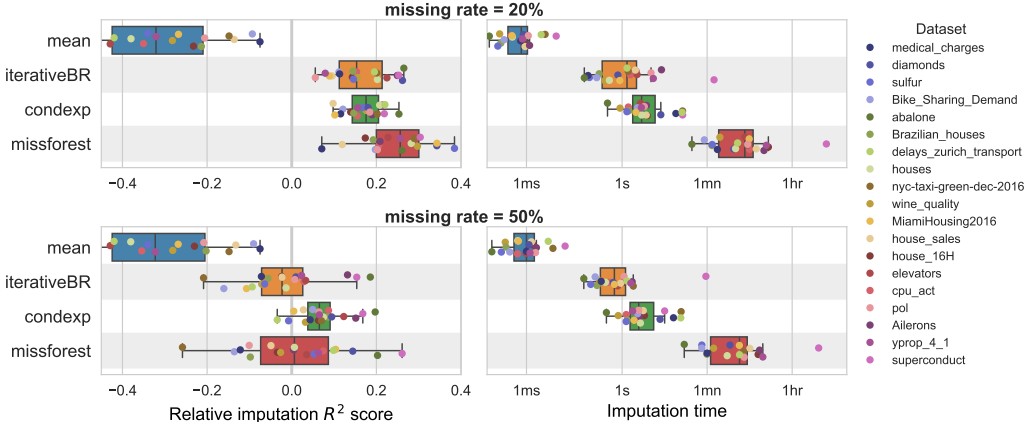

Figure 2: **Left: Imputer performance for recovery.** Performances are given as $R^2$ scores for each dataset relative to the mean performance across imputation techniques. A negative value indicates that a method perform worse than the average of other methods. **Right: Imputation time.**

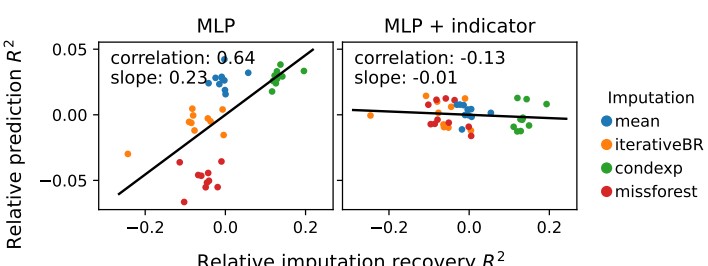

Figure 3: **Example fit of prediction performance as a function of imputation accuracy**, for the `Bike_Sharing_Demand` dataset and a missing rate of 50%: on the left using an MLP as predictor, and on the right an MLP with missingness indicator.

methods (an average difference of 0.5 $R^2$ points at 20% and 0.3 $R^2$ points at 50%) allows capturing differences in prediction performance.

## 4.3 LINKING IMPUTATION ACCURACY AND PREDICTION PERFORMANCES.

Combining the four imputation techniques with 10 repetitions of each experiment yields 40 (imputation $R^2$, prediction $R^2$) pairs for each model and dataset. To quantify how improvements in imputation accuracy translate into downstream prediction performance, we fit a linear regression using these 40 points for each model and dataset[1]. Figure 3 gives two examples of such fit: on the `Bike_Sharing_Demand` dataset, for a missing rate of 50%, the prediction $R^2$ increases as a function of the imputation $R^2$; the effect is greater for the MLP, for which the fit gives a slope of 0.23, than for the MLP with indicator for which the slope is -0.01.

Figure 4 summarizes the slopes estimated using the aforementioned methodology across all datasets, predictors with and without the indicator, and varying missing rates. Firstly, the fact that most slopes are positive indicates that better imputations correlate with better predictions, aligning with common beliefs. However, this observation should be interpreted with nuance in light of the effect sizes.

**Gains in prediction $R^2$ are 10% or less of the gains in imputation $R^2$.** Figure 4 shows that the slopes are typically small, rarely exceeding $0.1$. This implies that an improvement of $0.1$ in imputation $R^2$ typically leads to an improvement in prediction $R^2$ that is 10 times smaller, i.e. a gain of $0.01$ in prediction $R^2$, or even less. For XGBoost, the average slope across datasets in rather close to 0.025 or less (even zero without the mask at 20% missing rate). Thus, an enhancement of 0.3 in imputation $R^2$, which represents the average difference between the best of the worst imputer in this scenario (mean vs condexp in fig. 2), implies a gain in prediction $R^2$ of only 0.0075.

---

[1]The repetition identifier is also used as a covariate in the linear regression to account for the effects of the various train/test splits on prediction performance.

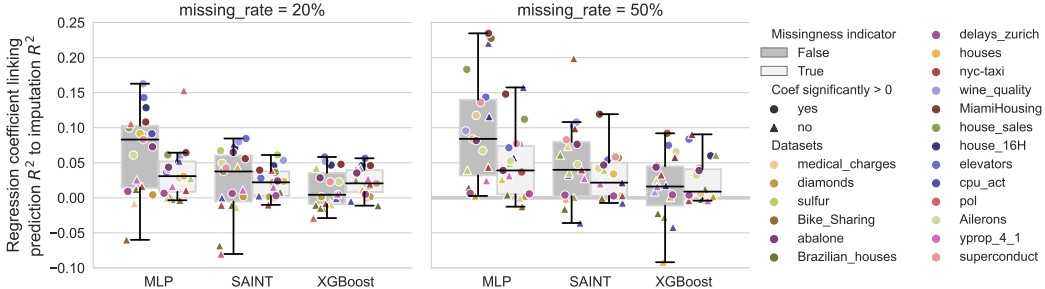

Figure 4: **Effect of the imputation recovery on the prediction performance.** We report the slope of the regression line where imputation quality is used to predict prediction performance. A coefficient is marked as significantly greater than zero (circle) if the associated p-value (one-sided T-test) is below 0.05 after Bonferroni correction for multiple testing.

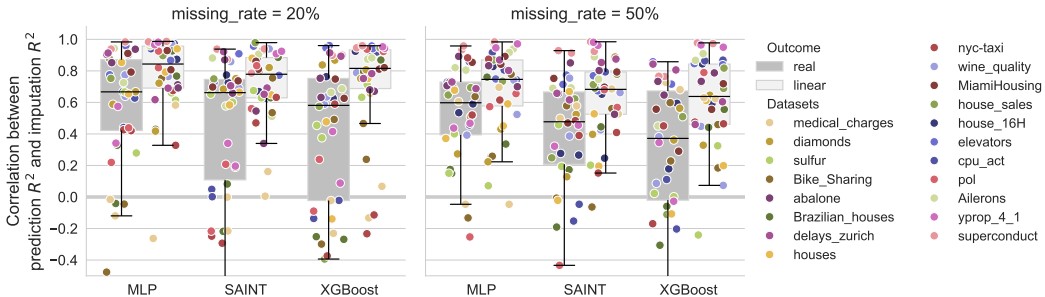

Figure 5: **Correlation between imputation quality and prediction performance.** A correlation close to 1 indicates that the quality of imputations is stronly associated to the quality of predictions, while a correlation close to zero means that the quality of predictions is not linked to the quality of imputations. Each correlation is computed using 40 different imputation/performance pairs, made of 4 imputation methods (mean, iterativeBR, missforest, condexp) repeated 10 times.

**Good imputations matter less for more expressive predictors.** Comparison between models shows a decrease in slope from MLP to SAINT, to XGBooost. A one-sided Wilcoxon signed-rank test assessing whether the median of each boxplot in Figure 4 is significantly greater than 0 reveals that the positive effect of imputation on prediction is significant for the MLP, but not for SAINT or XGBoost (case without missingness indicator). These results illustrate the idea that a powerful model can compensate for the simplicity or inaccuracy of an imputation (in our case, the MLP can be considered the least expressive model, and XGBoost the most expressive). Le Morvan et al. (2021) gives a formal proof in an extreme case: given enough samples, a sufficiently expressive model can always build a Bayes optimal predictor even on the simplest imputations (e.g. a constant).

**Good imputations matter less when adding the indicator.** Figure 4 shows that adding the missingness indicator clearly decreases the effect size: imputing better has less impact on performance when the indicator is used (we discuss this effect further in section 4.5).

**Good imputations matter less when the response is non-linear.** When the response $y$ is a linear function of the input $X$, the best predictor can be built using a linear model on the most accurate simple imputation. However, when the responses are non-linear, it may be difficult to learn the best possible predictor as it becomes a discontinuous function even with the most accurate imputation (Le Morvan et al., 2021). There are thus reasons to believe that response non-linearity, which is common in real data, alters the relationship between imputation accuracy and prediction performance. To investigate this, we compare the real datasets with matching semi-simulated datasets where $y$ is simulated as a linear function of the input $X$. We also measure correlation[2] (fig. 5) in addition to the slope, to quantify the reliability of the association: correlation captures not only the effect size (slope) but also the amount of noise (appendix C recalls this classic result) in the relationship. While the effects are similar between real and linear outcomes (fig. 11 gives effects in the semi-simulated case), the correlation between imputation accuracy and prediction performance, averaged across all datasets, is systematically smaller for real outcomes than for linear ones (fig. 5). The average decrease in correlation lies between 0.1 and 0.3 across models. Moreover, the variance in correlations for real outcomes is much larger, with many datasets with a near-zero correlation. This shows that the gains expected in prediction from better imputation are much more reliably achieved when the response is linear.

## 4.4 LOWER EFFECT OF IMPUTATION ACCURACY ON PREDICTION PERFORMANCE WITH MNAR MISSINGNESS.

We argue that MCAR provides a best-case scenario to study the potential benefit of imputation on prediction. It is indeed the easiest setting for imputation, as the missingness does not depend on the data. In addition, expecting general conclusions to apply universally across all MAR and MNAR

---

[2]Specifically we use the partial correlation, partialing out the effect of repeated train/test splits.

mechanisms is unrealistic, as these categories encompass families of missingness mechanisms with infinitely many variations (see, for example, Pereira et al. (2024)). Instead, the effects will depend on the specific MAR or MNAR model used. For instance, in self-censoring (a MNAR mechanism), the probability of a feature being missing can follow a hard-thresholding function based on underlying values, making imputation very hard, or an almost flat function, which is much closer to an easy MCAR scenario.

To illustrate the difference between MCAR and MNAR scenarios, we re-ran all experiments with a self-censoring mechanism, where the probability of missingness increases smoothly from 0 to 1 over the support of the data, according to a probit function (details in Appendix L.1). This mechanism was chosen to be neither too hard (e.g. hard thresholding), nor too easy (e.g almost MCAR). Figure 34 shows that the estimated effects are consistently lower than in the MCAR case. When the missingness indicator is not used, improving imputation quality even has, on average, a negative effect on prediction accuracy of XGBoost and SAINT. This is mainly because mean imputation performs well compared to more advanced strategies, likely because the mean allows to retain the information that a value was imputed. Overall, these experiments show that MCAR is a best-case scenario and that imputation in other settings will bring less benefits.

### 4.5    Why is the indicator beneficial, even with MCAR data?

In general, we find that adding the missingness indicator improves prediction. While it is expected that adding the indicator is beneficial in MNAR scenarios, as the missingness is informative, it is less obvious in the MCAR settings studied here. Indeed, the indicator contains absolutely no relevant information for predicting the outcome. To the best of our knowledge, the benefit of using an indicator in MCAR has not yet been established.

A possible theoretical explanation for this finding lies in the challenge of learning optimal prediction functions on imputed data. The best possible predictor in the presence of missingness can always be expressed as the composition of an imputation and a prediction function (Le Morvan et al., 2021). But, in general, the best prediction function on the imputed data can be challenging to learn, even for perfect conditional imputation. In fact, it often displays discontinuities on imputed points. We hypothesize that adding the missingness indicator simplifies modeling functions that exhibit discontinuities at these points, as the indicator can act as a switch to encode these discontinuities.

To assess the role of the information encoded in the missingness indicator, we repeated the experiments with a shuffled missingness indicator, where the columns of the indicator were randomly shuffled for each sample. This preserves the total number of missing values per sample but removes information about which specific features are missing. Figure 15a demonstrates that using a shuffled missingness indicator harms prediction performance, except for XGBoost for which performances are unchanged. In contrast, the true missingness indicator improves performances (fig. 1). Furthermore, the shuffled indicator does not affect the relationship between imputation accuracy and prediction accuracy (fig. 15b), whereas the true indicator reduces the effect size (fig. 4). These results confirm that the benefit of the missingness indicator is not due to a regularization or merely encoding the number of missing values. Experiments on feature importance further show that the importance of a feature drops when imputed compared to when observed (Appendix H), suggesting that imputations do not contribute to predictions as effectively as observed values.

The case of XGBoost in Figure 1 illustrates the importance of keeping the missingness information encoded. For 50% missing rate, in the absence of an indicator, no imputation benefits prediction with XGBoost, and the best option is to use the native handling of missing values. This suggests that XGBoost benefits from knowing which values are missing. With advanced imputations, distinguishing between imputed and observed values becomes challenging. Appending the indicator to the imputed data reinstates the missingness information unambiguously, which enables XGBoost to benefit from more advanced imputations, in particular missforest.

## 5    Conclusion

**Imputation matters for prediction, but only marginally.**    Prior theoretical work showed that in extreme cases (asymptotics), imputation does not matter for predicting with missing values. We quantified empirically the effect of imputation accuracy gains on prediction performance across

many datasets and scenarios. We show that in practice, imputation does play a role. But various factors modulate the importance of better imputations for prediction: investing in better imputations will be less beneficial when a flexible model is used, when a missing-value indicator is used, and if the response is non-linear. These results are actually in line with the theoretical results suggesting that imputation does not matter, as these hold for very flexible models (ie universally consistent). A notable new insight is that adding a missing-value indicator as input is beneficial for prediction performances *even for MCAR settings*, where missingness is uninformative.

We show that large gains in imputation accuracy translate into small gains in prediction performance. These results were drawn from a favorable MCAR setting, and it is likely that with native missingness, often Missing Non At Random (MNAR), the performance gains are even smaller. As novel imputation methods usually provide small gains in imputation accuracy compared to the state-of-the-art, the corresponding gains in downstream prediction tasks are likely to be even smaller.

There are multiple potential reasons why imputation gains do not always correlate with performance gains. For instance, some features may be well recovered, but not useful in the prediction because they are not predictive. Or even with accurate imputations, it may still be difficult to learn a predictor that performs well for all missing data patterns (Le Morvan et al., 2021). Finally, the imputation accuracy is also probably an imperfect measure of the potential gains in prediction: in our experiments on 50% missing rate, missforest and iterativeBR performs comparable on average yet missforest-based predictors tend to outperform those based on iterativeBR.

**Limitations and future work.** It would be valuable to investigate whether imputations based on random draws outperform deterministic imputations for downstream prediction tasks, and the usefulness of multiple imputations (Perez-Lebel et al., 2022). A related question is whether reconstructing well the data distribution is important for better predictions. Shadbahr et al. (2023) shows that it does not seem crucial for classification performances but may compromise more seriously model interpretability. Finally, appending the indicator to the input improves predictions, but doubles the feature count and may not be the most effective encoding for enhancing downstream performance. Alternative approaches, such as missingness-aware feature encodings (Lenz et al., 2024), learned missingness embeddings (Somepalli et al., 2021), or missingness-aware layers (Le Morvan et al., 2020), have been proposed, but further investigation is needed in these directions.

**Outlook** Improving imputation is often a difficult way of improving prediction. On top of imputation, future research could focus more on developing advanced modeling techniques that can inherently handle missing values and effectively incorporate missingness indicators to improve predictive performance.

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

# Appendix

## Table of Contents

## A LIST OF DATASETS.

This benchmark was created by Grinsztajn et al. (2022), and is available on OpenML at
`https://www.openml.org/search?type=benchmark&study_type=task&sort=`
`tasks_included&id=336`.

Table 1: **Dataset dimensions.**

| dataset | d | n_train | n_test |
|---|---|---|---|
| house_16H | 16 | 18185 | 2274 |
| cpu_act | 21 | 6553 | 820 |
| elevators | 16 | 13279 | 1661 |
| wine_quality | 11 | 5197 | 651 |
| Brazilian_houses | 8 | 8553 | 1070 |
| house_sales | 15 | 17290 | 2162 |
| sulfur | 6 | 8064 | 1009 |
| Ailerons | 33 | 11000 | 1375 |
| Bike_Sharing_Demand | 6 | 13903 | 1739 |
| diamonds | 6 | 43152 | 5394 |
| fifa | 5 | 14450 | 1807 |
| houses | 8 | 16512 | 2064 |
| medical_charges | 3 | 50000 | 50000 |
| MiamiHousing2016 | 13 | 11145 | 1394 |
| nyc-taxi-green-dec-2016 | 9 | 50000 | 50000 |
| pol | 26 | 12000 | 1500 |
| superconduct | 79 | 17010 | 2127 |
| yprop_4_1 | 42 | 7108 | 889 |
| abalone | 7 | 3341 | 419 |

## B HYPERPARAMETER SEARCH SPACES

Table 2: **XGBoost hyperparameter space.** We used the `XGBRegressor` from the `xgboost`
Python library. The hyperparameters optimized are commonly accepted as the most important ones.
The variation ranges are inspired by the ones used in Grinsztajn et al. (2022), while the default
hyperparameters are those of the `xgboost` library.

| parameter | range | log scale | default |
|---|---|---|---|
| n_estimators | $[100, 2000]$ | no | 100 |
| max_depth | $[1, 6]$ | no | 6 |
| learning_rate | $[10^{-5}, 0.7]$ | yes | 0.3 |
| reg_alpha | $[10^{-8}, 10^2]$ | yes | $10^{-8}$ |
| reg_lambda | $[1, 4]$ | yes | 1 |
| early_stopping_rounds | - | - | 20 |

Table 3: **MLP hyperparameter space.** We implemented the MLP in PyTorch. The parameter $d$ for the width of the MLP represents the number of features. When $d > 1024$, the width is taken equal to the number of features $d$.

|           | parameter | range | default |
|-----------|-----------|-------|---------|
| MLP       | depth     | $[\![0, 6]\!]$ | 3 |
|           | width     | $[d, \min(10d, 1024)]$ | 3d |
|           | dropout rate | $[0, 0.5]$ | 0.2 |
| Optimizer | name      | - | AdamW |
|           | weight decay | - | $10^{-6}$ |
|           | learning rate | - | $10^{-3}$ |
| Scheduler | name      | - | `ReduceLROnPlateau` |
|           | factor    | - | 0.2 |
|           | patience  | - | 10 |
|           | threshold | - | $10^{-4}$ |
| General   | max nb. epochs | - | 2000 |
|           | early stopping | - | Yes |
|           | batch size | - | 256 |

Table 4: **SAINT default hyperparameters.** We used the implementation provided by Somepalli et al. (2021). $d$ refers to the number of features of the dataset. We did not use a scheduler with SAINT. We followed the default configuration provided by the paper introducing SAINT (Somepalli et al., 2021) when there is both intersample and feature attention (i.e. attention_type = `'colrow'`).

|           | parameter | default |
|-----------|-----------|---------|
| SAINT     | dim       | 32 if $d < 70$, 16 if $d \in [70, 200]$, 4 if $d \geq 200$ |
|           | depth     | 1 |
|           | heads     | 4 |
|           | attn_dropout | 0.8 |
|           | ff_dropout | 0.8 |
|           | attentiontype | colrow |
| Optimizer | name      | AdamW |
|           | weight decay | $10^{-2}$ |
|           | learning rate | $10^{-4}$ |
| General   | max nb. epochs | 100 |
|           | early stopping | yes |
|           | batch size | 256 |

## C    LINK BETWEEN CORRELATION AND EFFECT SIZE.

For completeness, we recall below the relationship between correlation and effect size.

**Proposition C.1** (Link between correlation and effect size.)**.**  *Let $X_1 \in \mathbb{R}$ be a random variable, and $\beta \in \mathbb{R}$ a parameter. Furthermore, define:*

$$X_2 = \beta X_1 + \epsilon \quad where \quad \mathbb{E}[\epsilon | X_1] = 0, \ var(\epsilon) = \sigma^2.$$

*Then:*

$$cor(X_1, X_2) = \frac{1}{\sqrt{1 + \frac{\sigma^2}{\beta^2 var(X_1)}}}$$

*Proof.*  Let's first derive the expression of the variance of $X_2$:

$$\begin{aligned}
\mathrm{Var}(X_2) &= \mathbb{E}\left[(X_2 - \mathbb{E}[X_2])^2\right] \\
&= \mathbb{E}\left[(\beta X_1 + \epsilon - \beta \mathbb{E}[X_1])^2\right] \\
&= \mathbb{E}\left[(\beta(X_1 - \mathbb{E}[X_1]))^2 + \epsilon^2\right] \\
&= \beta^2 \mathrm{var}(X_1) + \sigma^2
\end{aligned}$$

It follows that:

$$\begin{aligned}
\mathrm{cor}(X_2, X_1) &= \frac{\mathbb{E}[(X_2 - \mathbb{E}[X_2])(X_1 - \mathbb{E}[X_1])]}{\sqrt{\mathrm{var}(X_1)\,\mathrm{var}(X_2)}} \\
&= \frac{\mathbb{E}\left[\beta(X_1 - \mathbb{E}[X_1])^2\right]}{\sqrt{\mathrm{var}(X_1)\,\mathrm{var}(X_2)}} \\
&= \beta \frac{\sqrt{\mathrm{var}(X_1)}}{\sqrt{\mathrm{var}(X_2)}} \\
&= \beta \frac{\sqrt{\mathrm{var}(X_1)}}{\sqrt{\beta^2 \mathrm{var}(X_1) + \sigma^2}} \\
&= \frac{1}{\sqrt{1 + \frac{\sigma^2}{\beta^2 \mathrm{var}(X_1)}}}
\end{aligned}$$

$\square$

In this work, we look at the effect of imputation accuracy ($X_1$) on prediction performance ($X_2$). Hence, in a case where the imputation accuracy $X_1$ covers a wider range of values, i.e., var($X_1$) is larger, but the effect $\beta$ and the noise $\sigma^2$ stay the same, then the correlation between imputation accuracy and prediction performance increases.

## D    CRITICAL DIFFERENCE DIAGRAMS.

Figures 6 to 9 give the **Critical Difference diagrams across all predictors and imputers** of average score ranks for a significance level of 0.05. The difference in ranks for all methods covered by the same black crossbar are not statistically significant according to a Nemenyi test for multiple pairwise comparisons. The colors encode the imputation type, the markers identify the model, and the line types encode the presence or absence of an indicator.

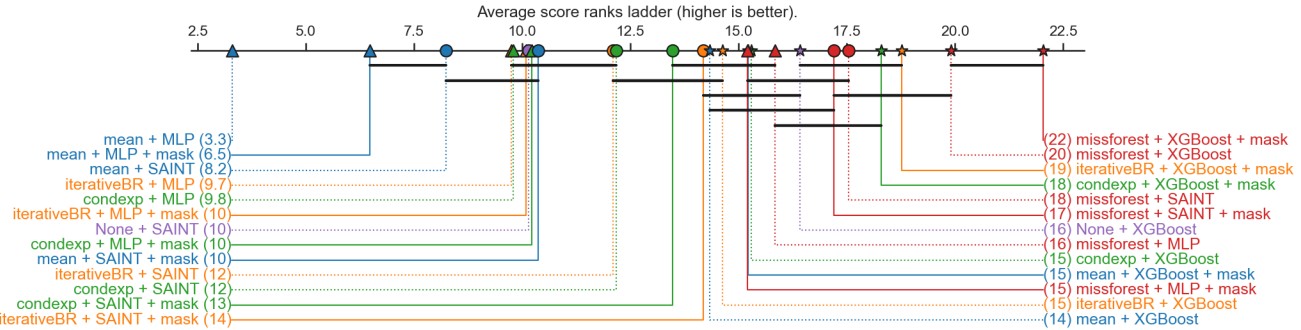

Figure 6: Critical Difference diagram - 20% missingness rate.

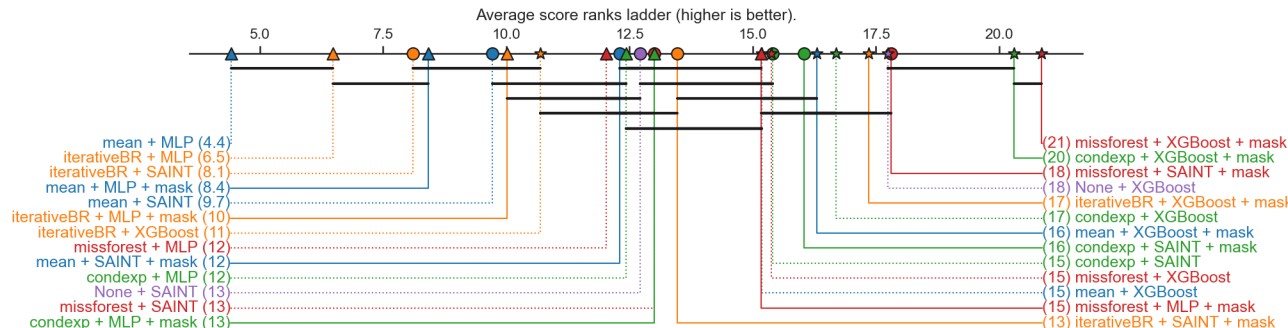

Figure 7: Critical Difference diagram - 50% missingness rate.

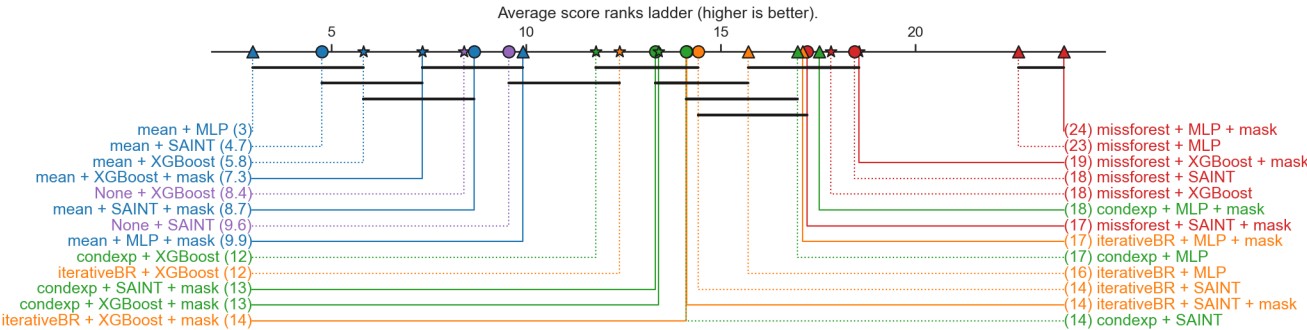

Figure 8: Critical Difference diagram - 20% missingness rate, semi-synthetic data with linear outcomes.

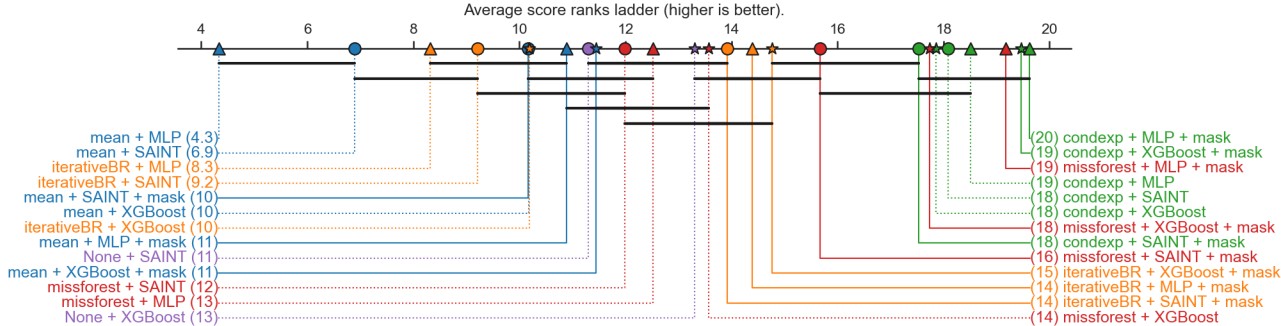

Figure 9: Critical Difference diagram - 50% missingness rate, semi-synthetic data with linear outcomes.

# E  PREDICTION PERFORMANCES FOR THE SEMI-SYNTHETIC DATA.

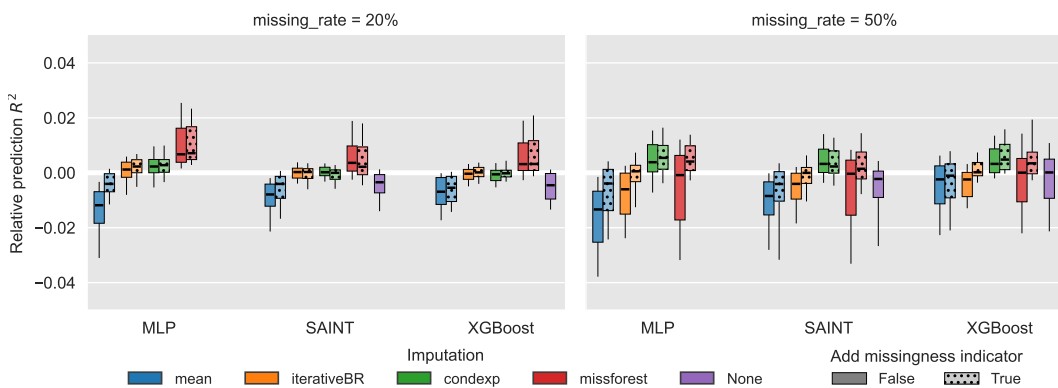

Figure 10: **Relative prediction performances for the semi-synthetic data with linear outcomes across datasets for different imputations, predictors, and use of the missingness indicator.** Each boxplot represents 200 points (20 datasets with 10 repetitions per dataset). The performances shown are $R^2$ scores on the test set relative to the mean performance across all models for a given dataset and repetition. A value of 0.01 indicates that a given method outperforms the average performance on a given dataset by 0.01 on the $R^2$ score.

# F  REGRESSION SLOPES FOR THE SEMI-SYNTHETIC DATA.

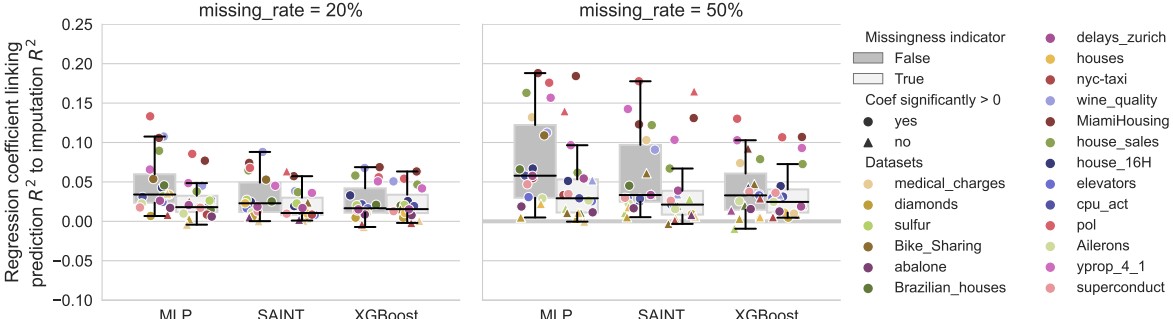

Figure 11: **Effect of the imputation recovery on the prediction performance** for the semi-synthetic data with linear outcomes. We report the slope of the regression line where imputation quality is used to predict prediction performance.

## G EFFECT OF THE MISSING RATE.

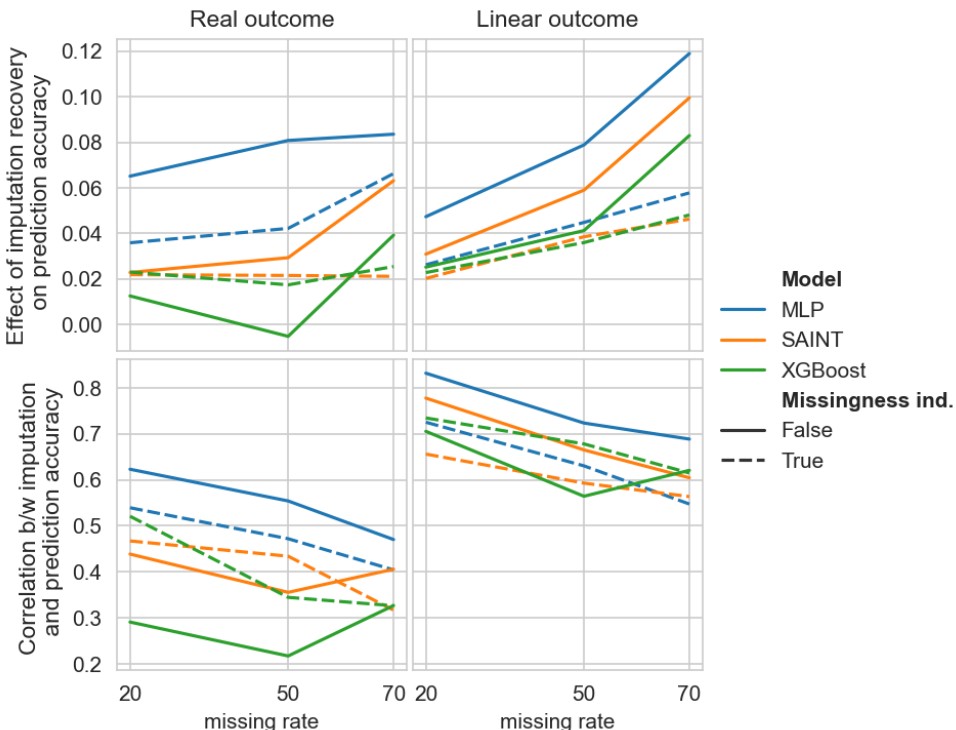

Figure 12: **Increasing missing rates lead to larger effects (slopes) but noisier associations (lower correlation)**. The values reported are median across datasets, for each model. Data is MCAR.

Figure 12 shows that effects are larger for higher missing rates: this is particularly clear for linear outcomes, but less for real outcomes. This suggests that imputation matters more at higher missing rates, although for real outcomes and powerful models, these effects are still very small. By contrast, correlations decrease when the missing rate increases, i.e, the association is noisier (less likely to be significant).

# H   FEATURE IMPORTANCE DEPENDING ON THE MISSINGNESS STATUS.

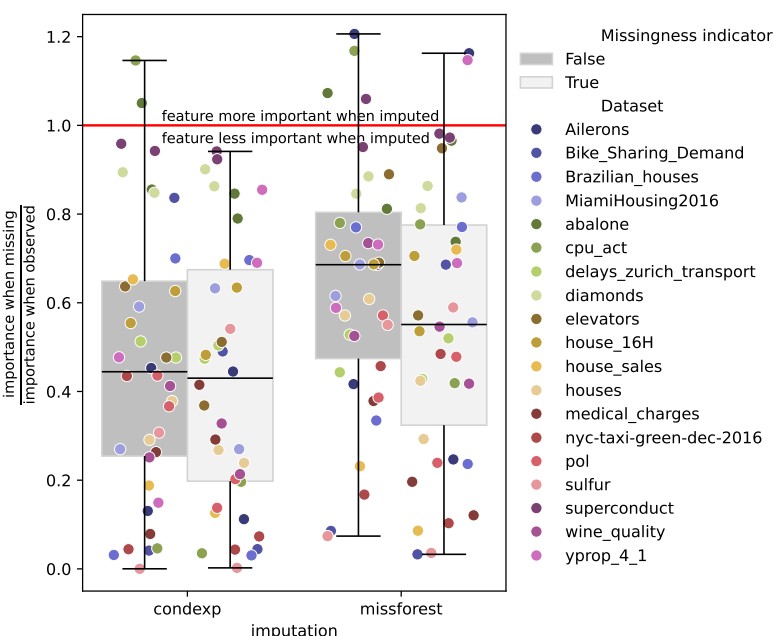

Figure 13: **Ratio of importances (when missing over when observed) for the two most important features of each dataset.** Importances are calculated with feature permutation. The 2 most important features per dataset are identified based on the whole test set. For each feature j, a permutation importance is then calculated based on the subset of test samples where feature $j$ is missing, and the subset where it is observed. The ratio between these two values is then reported on the figure, where a point refers to one feature, and its color identifies the dataset it belongs to. A ratio of 1 indicates that the feature is as important whether it is imputed or observed (red line). A ratio of 0.1 means that it is 10 times less important when it is imputed compared to when it is observed.

This experiment was conducted using XGBoost with condexp or missforest imputation, both with and without the mask. Figure 13 indicates that on average, a feature is half as important when imputed compared to when observed, with considerable variability (i.e., many features are 10 times less important when imputed, and some features remain as important when imputed). When a mask is used, importances drop significantly more with missforest imputation compared to when no mask is used (Wilcoxon signed-rank test p-value < 0.01). However, this effect is not significant with condexp imputation.

# I INVESTIGATING THE ROLE OF THE MISSINGNESS INDICATOR.

## I.1 PREDICTION PERFORMANCE GAINS WHEN USING THE MISSINGNESS INDICATOR VERSUS IMPUTATION ACCURACY

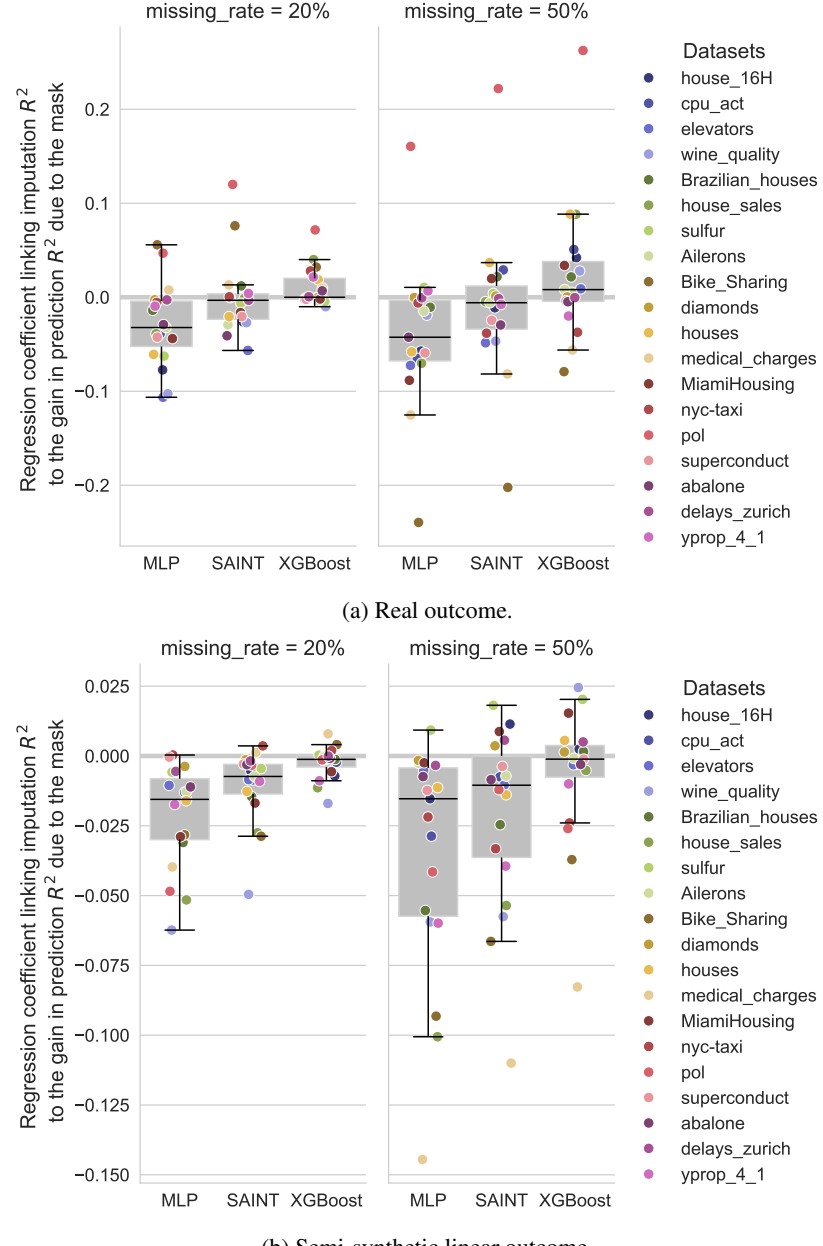

(a) Real outcome.

(b) Semi-synthetic linear outcome.

Figure 14: **Effect of imputation accuracy on the improvement in prediction when using the mask, compared to not using it.**

Most effects are negative, indicating that using the missingness indicator brings the largest boost in prediction performance when imputations have low accuracy. Moreover, effects are strongest for the MLP and smallest for XGBoost, meaning that with more powerful models, prediction boosts due to the missingness indicator are less pronounced.

### I.2 SHUFFLING THE MISSINGNESS INDICATOR.

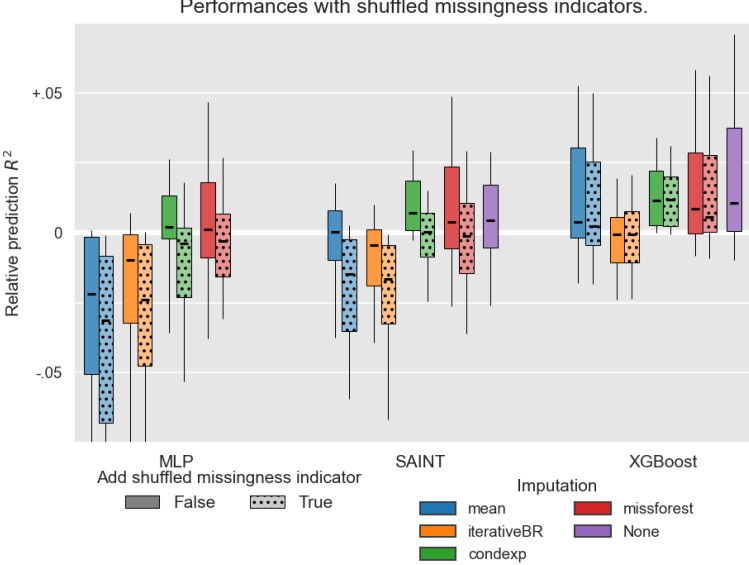

(a) Effect of appending a shuffled mask on prediction performances. Real outcome, 50% MCAR missingness.

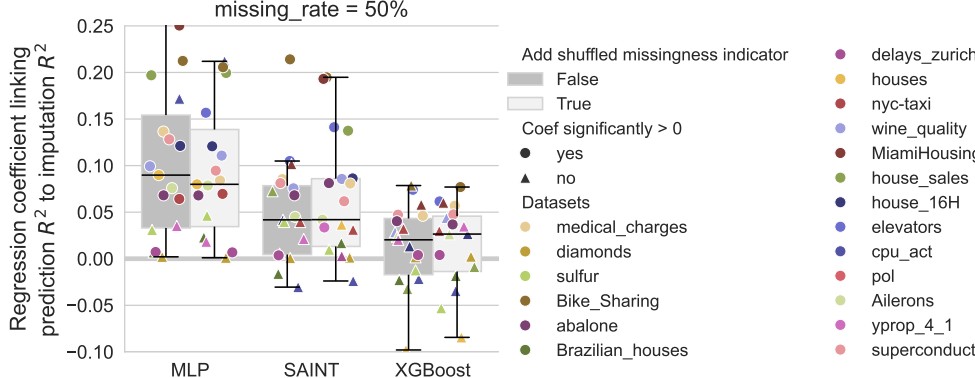

(b) Comparing effect sizes of imputation accuracy on prediction accuracy, with a shuffled mask versus without mask. Real outcome, 50% MCAR missingness.

Figure 15: **Effects of appending a shuffled mask**

We repeat experiments with a shuffled missingness indicator, where the columns of the indicator are shuffled for each sample. This preserves the total number of missing values per sample but removes information about which specific features are missing.

Figure 15a demonstrates that using a shuffled missingness indicator harms prediction performance, except for XGBoost for which performances are unchanged. In contrast, the true missingness indicator improves performances (fig. 1). Furthermore, the shuffled indicator does not affect the relationship between imputation accuracy and prediction accuracy (fig. 15b), whereas the true indicator reduces the effect size (fig. 4).

These results confirm that the benefit of the missingness indicator is not due to a regularization or merely encoding the number of missing values. Prediction models effectively leverage information about which features are missing, even though under MCAR, this information is unrelated to the unobserved values.

# J   COMPUTATION TIMES PER METHOD.

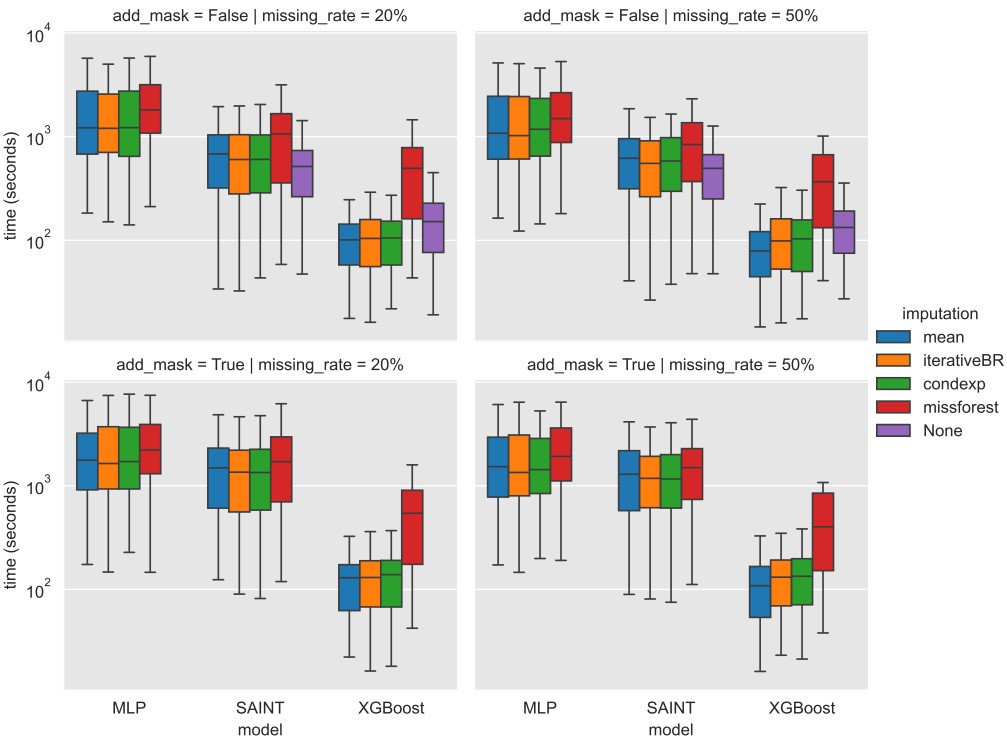

Figure 16: **Running time for each model**, including the 50 iterations of hyperparameter search for XGBoost and MLP.

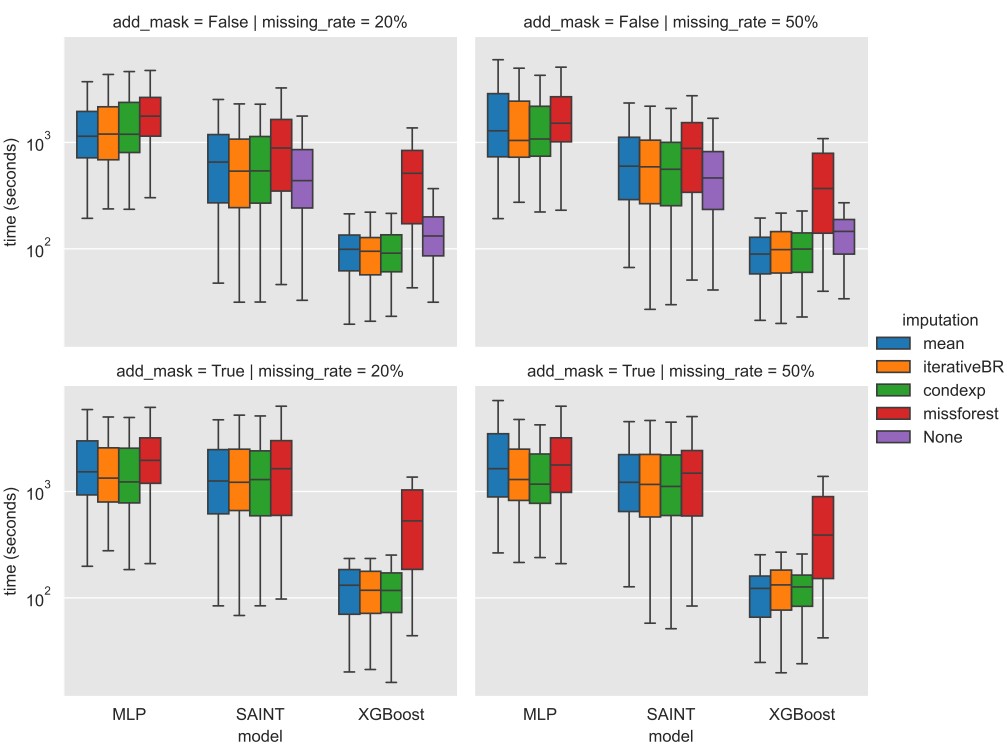

Figure 17: **Running time for each model** for the semi-synthetic data with linear outcomes, including the 50 iterations of hyperparameter search for XGBoost and MLP.

# K    SCATTERPLOTS OF PREDICTION $R^2$ VS IMPUTATION $R^2$ FOR EACH MODEL AND DATASET.

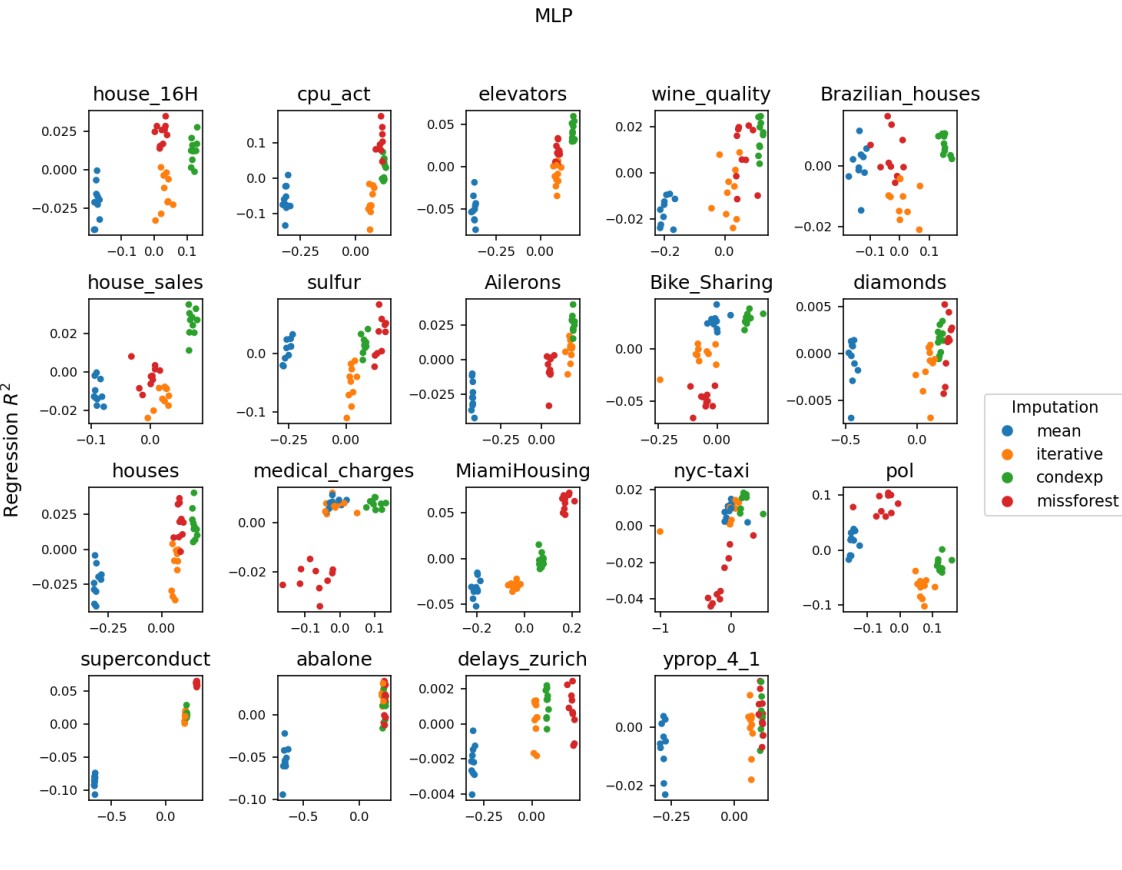

Figure 18: **Prediction $R^2$ vs imputation $R^2$ for a MLP** - missing rate 50%. The $R^2$ scores are given relative to the mean $R^2$ score, with the effects of experiment repetitions eliminated (i.e. the effect of the train/test splits on the performance)

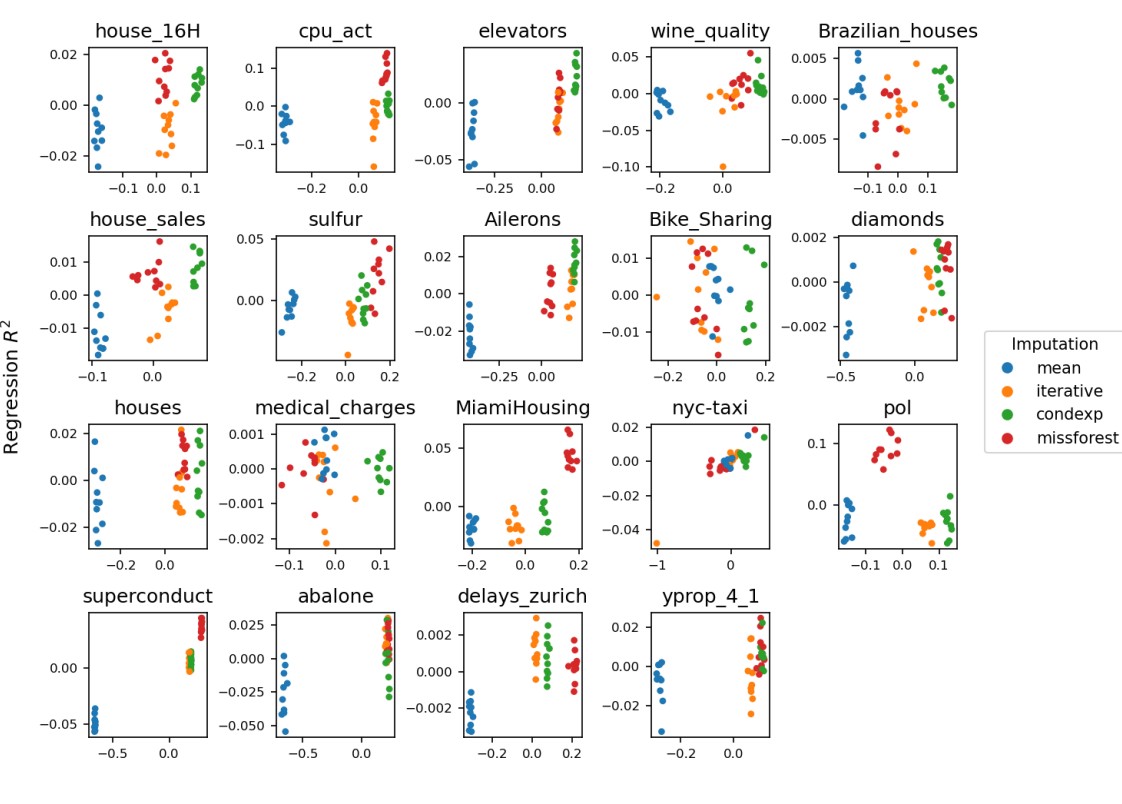

Figure 19: **Prediction $R^2$ vs imputation $R^2$ for a MLP + indicator** - missing rate 50%. The $R^2$ scores are given relative to the mean $R^2$ score, with the effects of experiment repetitions eliminated (i.e. the effect of the train/test splits on the performance)

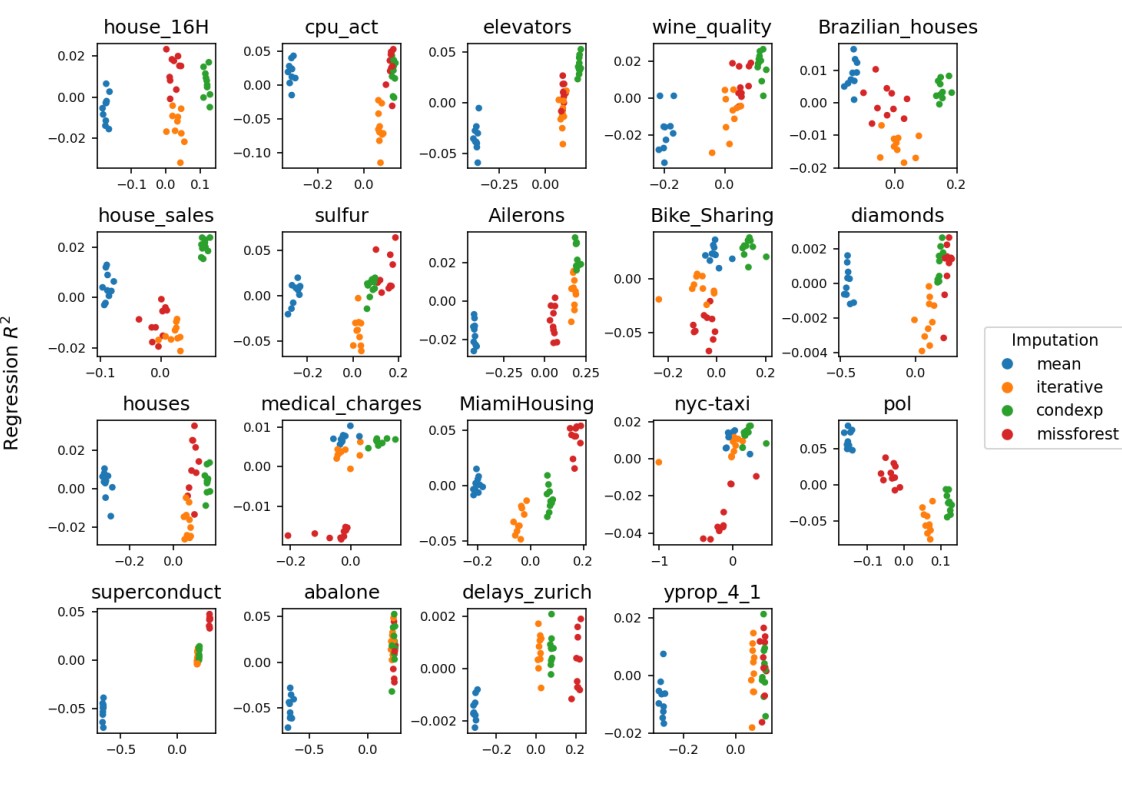

Figure 20: **Prediction $R^2$ vs imputation $R^2$ for SAINT** - missing rate 50%. The $R^2$ scores are given relative to the mean $R^2$ score, with the effects of experiment repetitions eliminated (i.e. the effect of the train/test splits on the performance)

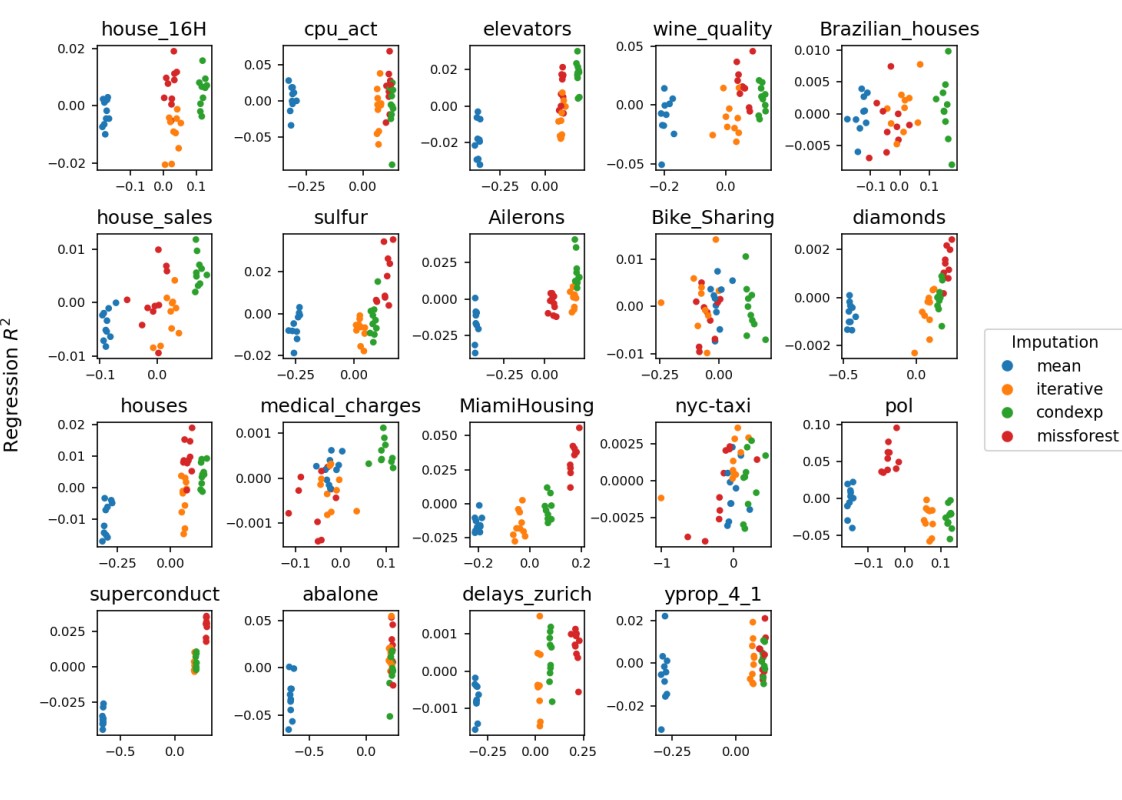

Figure 21: **Prediction $R^2$ vs imputation $R^2$ for SAINT + indicator** - missing rate 50%. The $R^2$ scores are given relative to the mean $R^2$ score, with the effects of experiment repetitions eliminated (i.e. the effect of the train/test splits on the performance)

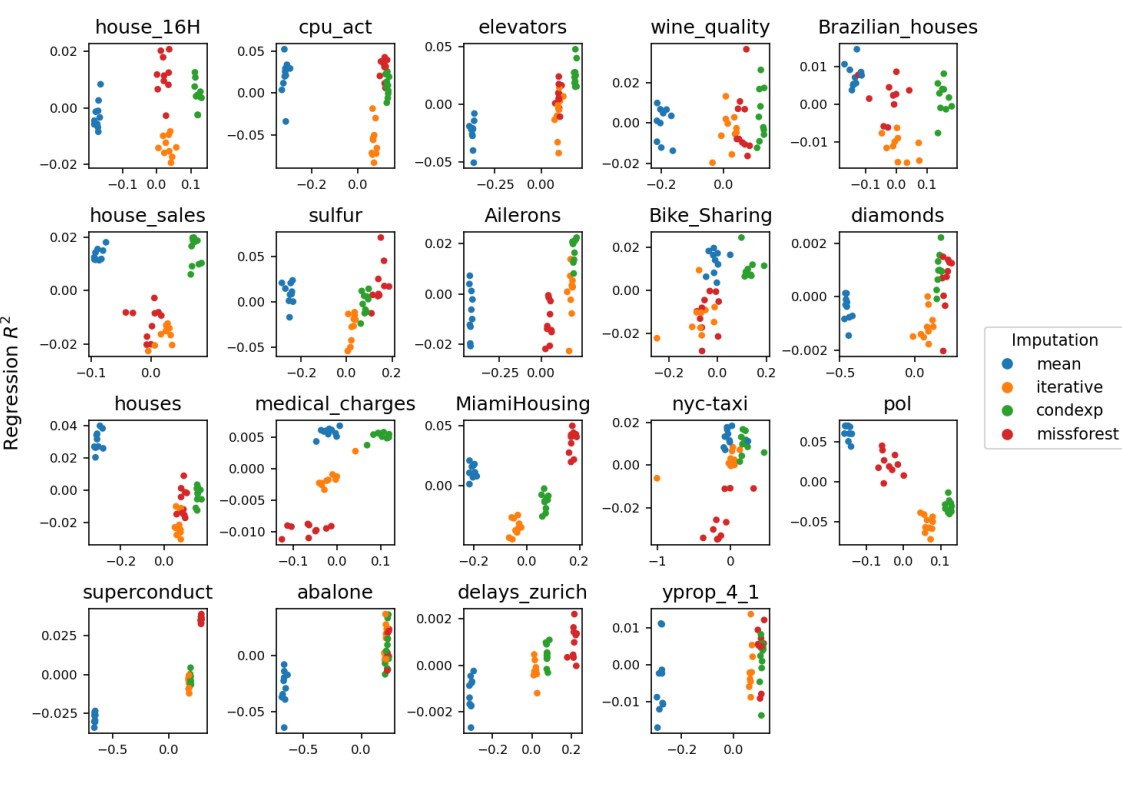

Figure 22: **Prediction $R^2$ vs imputation $R^2$ for XGBoost** - missing rate 50%. The $R^2$ scores are given relative to the mean $R^2$ score, with the effects of experiment repetitions eliminated (i.e. the effect of the train/test splits on the performance)

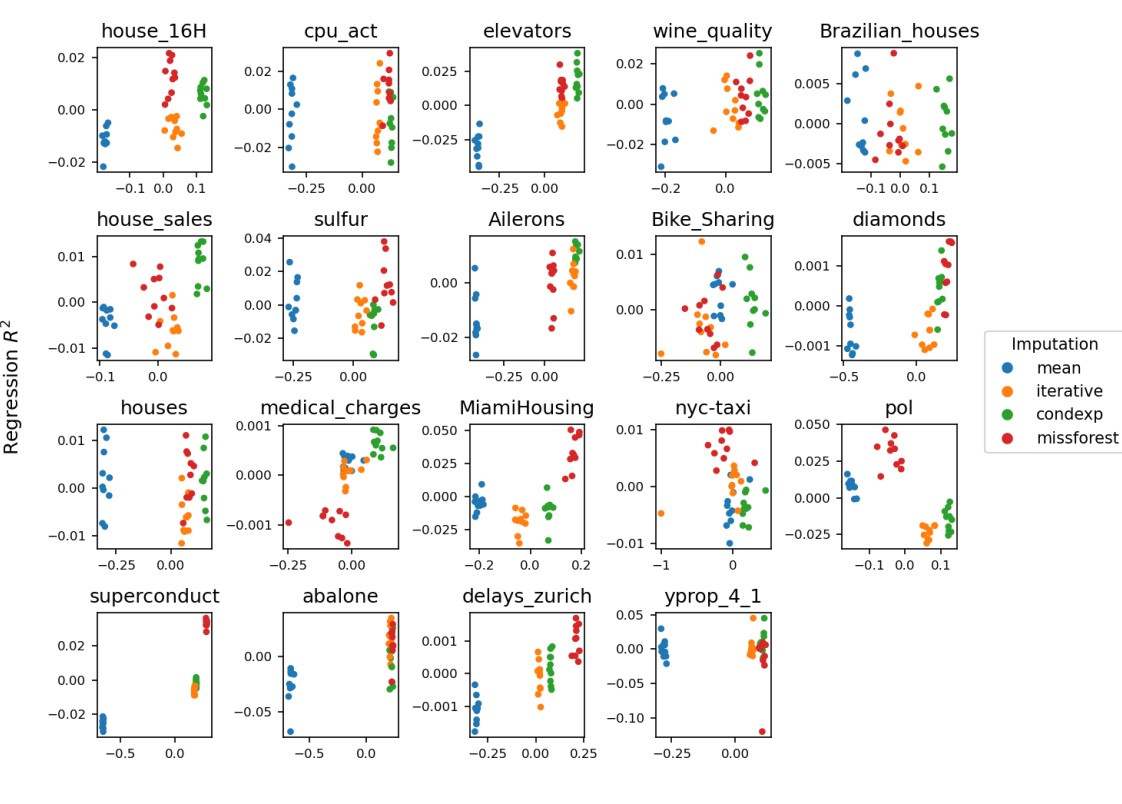

Figure 23: **Prediction $R^2$ vs imputation $R^2$ for XGBoost + indicator** - missing rate 50%. The $R^2$ scores are given relative to the mean $R^2$ score, with the effects of experiment repetitions eliminated (i.e. the effect of the train/test splits on the performance)

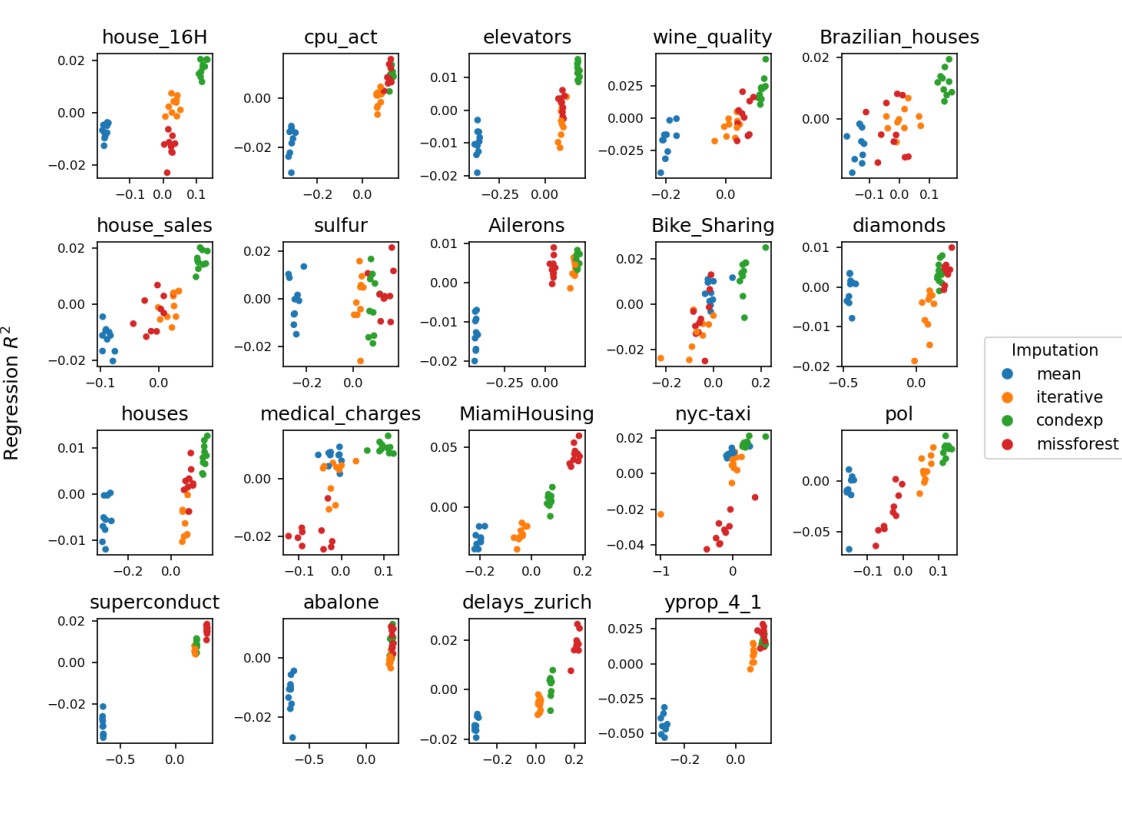

Figure 24: **Prediction $R^2$ vs imputation $R^2$ for a MLP** - semi-synthetic data with linear outcomes, missing rate 50%. The $R^2$ scores are given relative to the mean $R^2$ score, with the effects of experiment repetitions eliminated (i.e. the effect of the train/test splits on the performance)

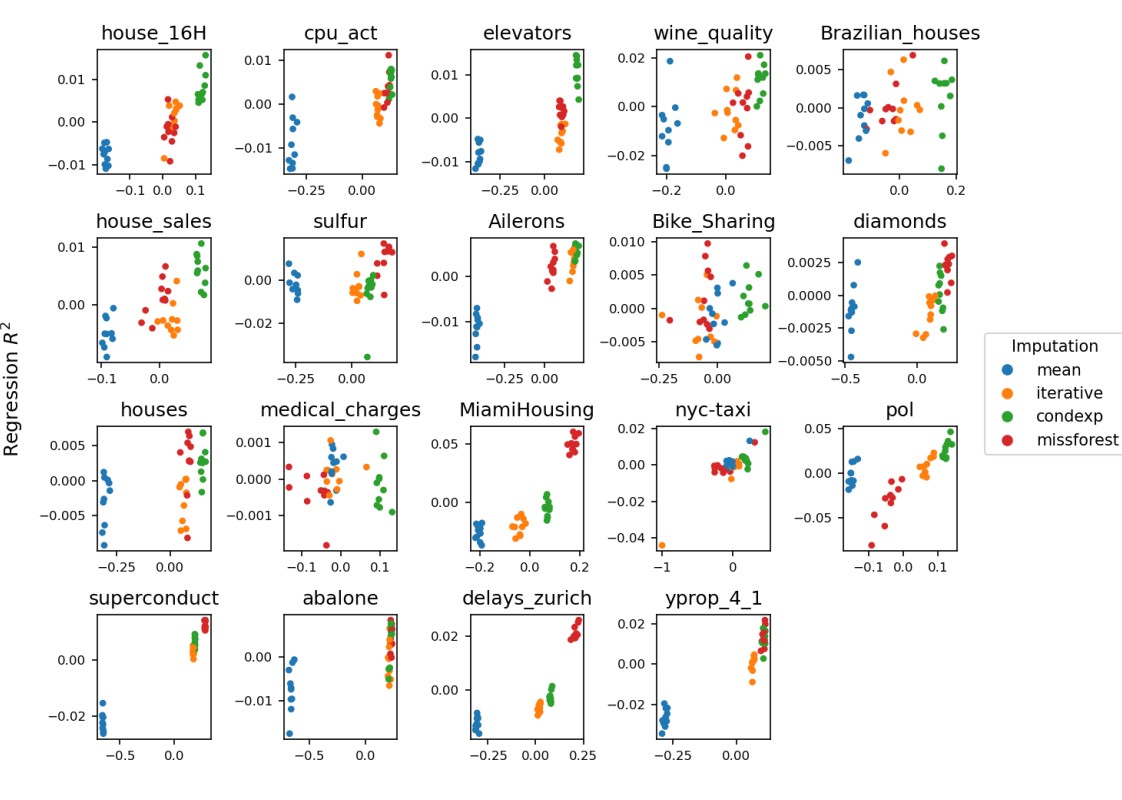

Figure 25: **Prediction** $R^2$ **vs imputation** $R^2$ **for a MLP + indicator** - semi-synthetic data with linear outcomes, missing rate 50%. The $R^2$ scores are given relative to the mean $R^2$ score, with the effects of experiment repetitions eliminated (i.e. the effect of the train/test splits on the performance)

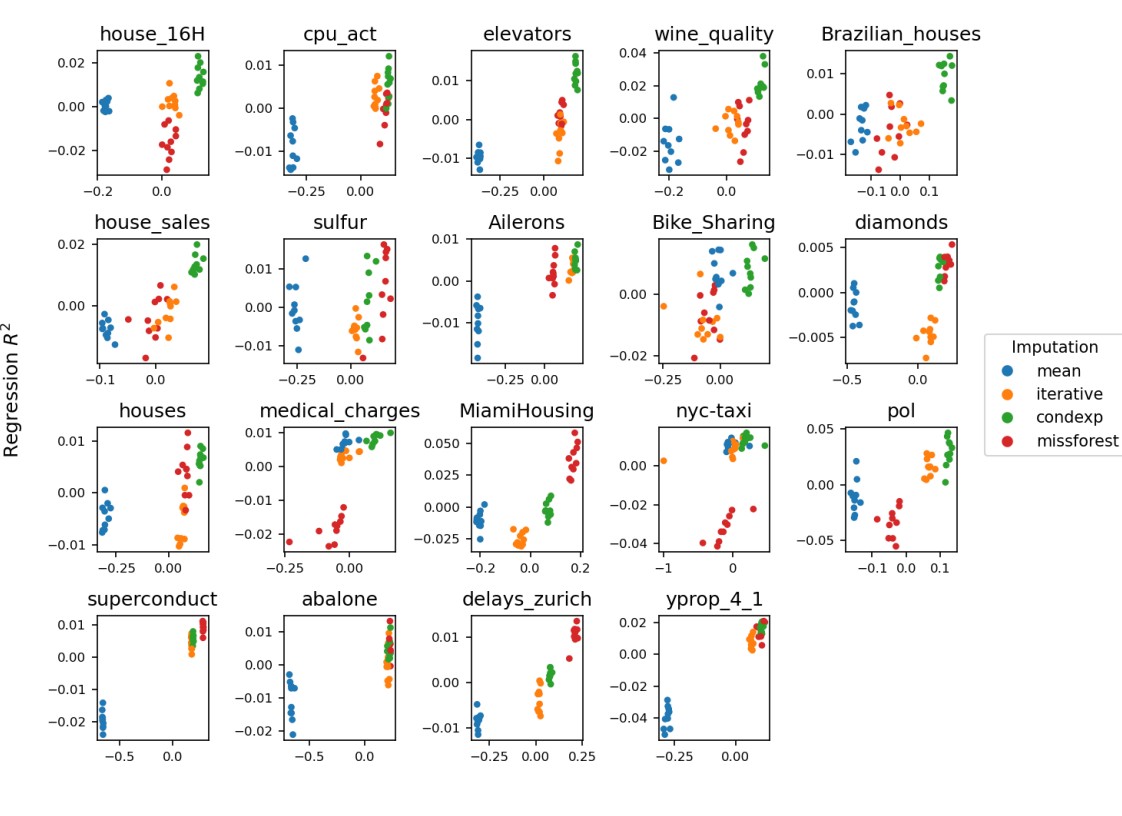

Figure 26: **Prediction $R^2$ vs imputation $R^2$ for SAINT** - semi-synthetic data with linear outcomes, missing rate 50%. The $R^2$ scores are given relative to the mean $R^2$ score, with the effects of experiment repetitions eliminated (i.e. the effect of the train/test splits on the performance)

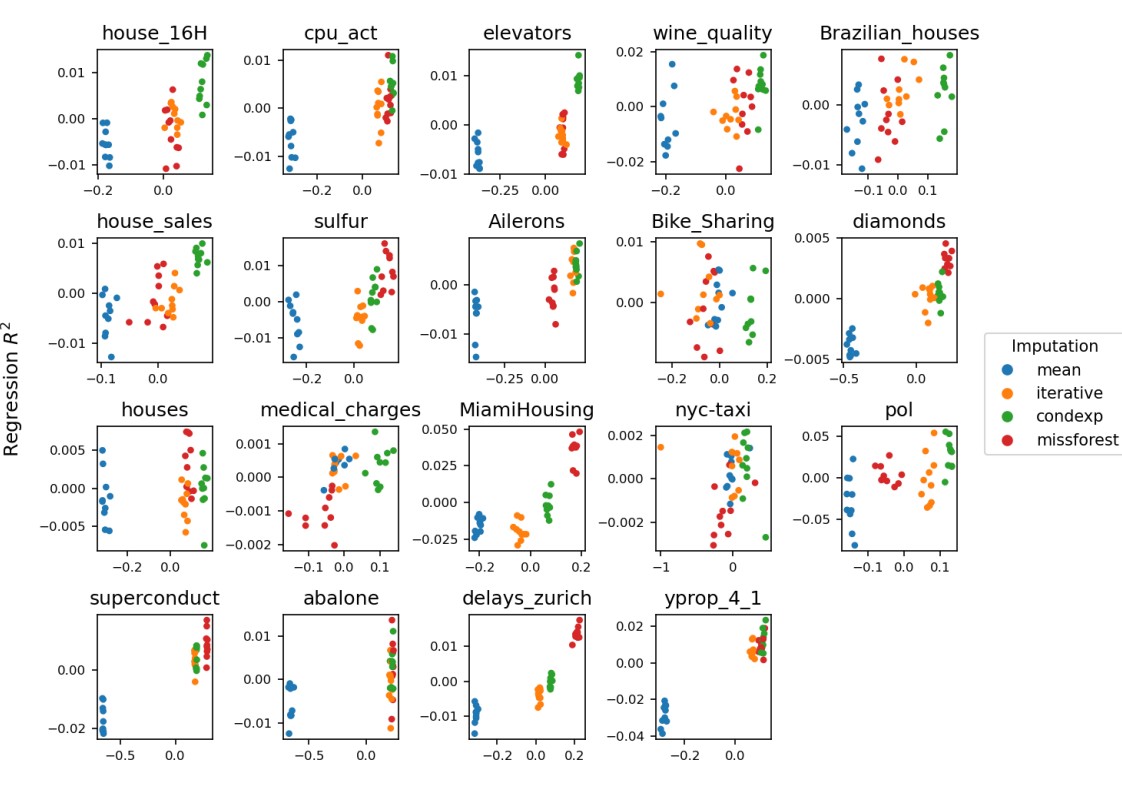

Figure 27: **Prediction** $R^2$ **vs imputation** $R^2$ **for SAINT + indicator** - semi-synthetic data with linear outcomes, missing rate 50%. The $R^2$ scores are given relative to the mean $R^2$ score, with the effects of experiment repetitions eliminated (i.e. the effect of the train/test splits on the performance)

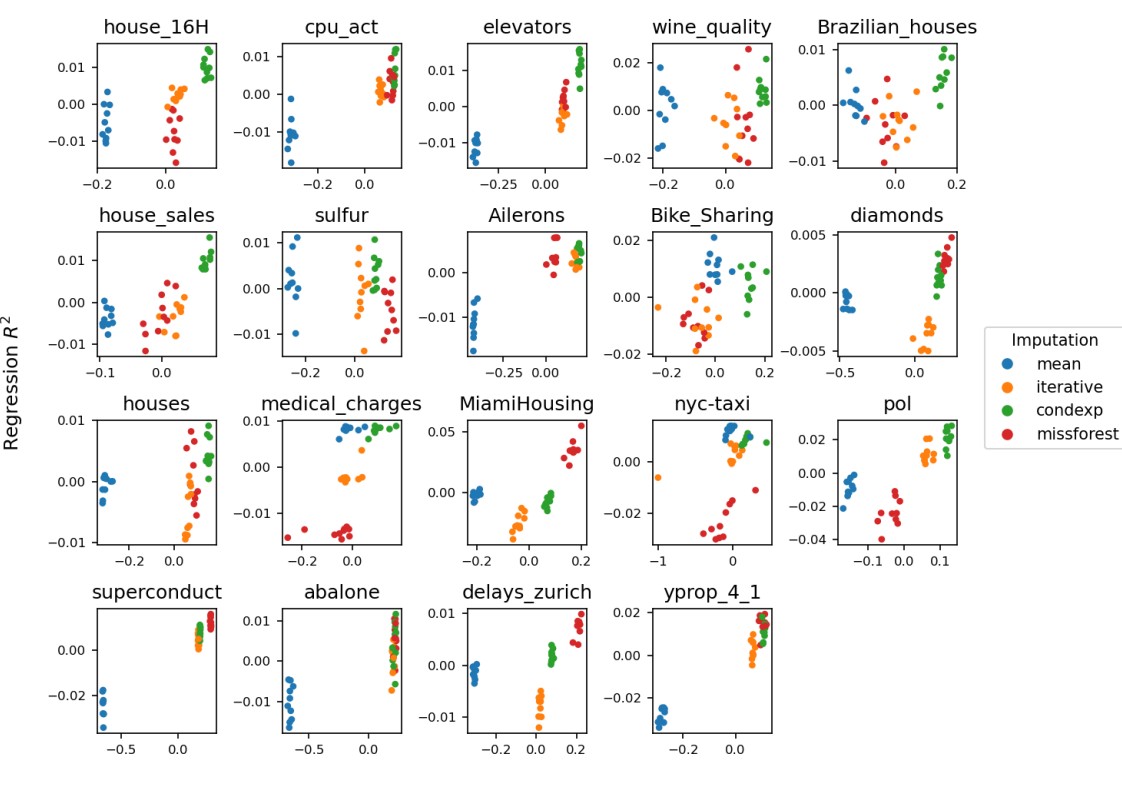

Figure 28: **Prediction** $R^2$ **vs imputation** $R^2$ **for XGBoost** - semi-synthetic data with linear outcomes, missing rate 50%. The $R^2$ scores are given relative to the mean $R^2$ score, with the effects of experiment repetitions eliminated (i.e. the effect of the train/test splits on the performance)

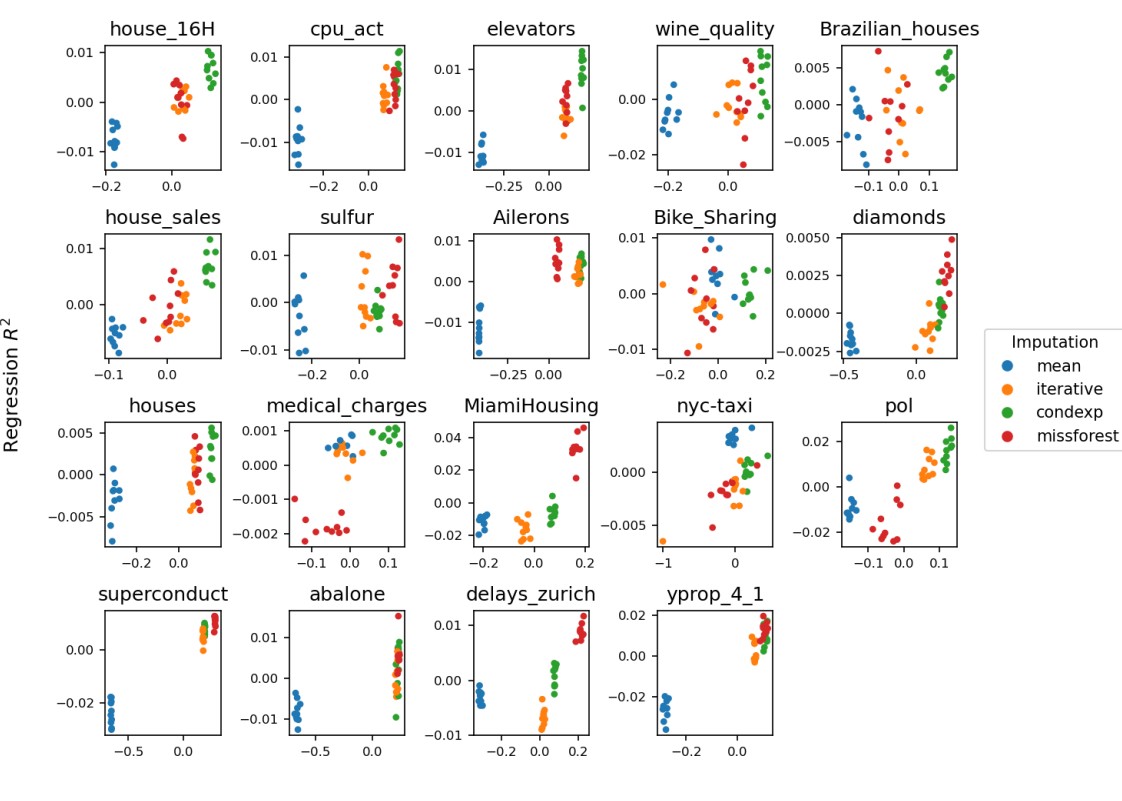

Figure 29: **Prediction** $R^2$ **vs imputation** $R^2$ **for XGBoost + indicator** - semi-synthetic data with linear outcomes, missing rate 50%. The $R^2$ scores are given relative to the mean $R^2$ score, with the effects of experiment repetitions eliminated (i.e. the effect of the train/test splits on the performance)

# L  MNAR SCENARIO.

## L.1  DESCRIPTION OF THE SELF-CENSORING MECHANISM.

The MNAR mechanism implemented is a probit self-masking, defined for any feature $j$ as:

$$P(M_j = 1|X_j) = \Phi(\lambda_j X_j - c_j)$$

where $\Phi$ denotes the probit function and $\lambda_j \in \mathbb{R}$, $c_j \in \mathbb{R}$ its slope and bias.

Denoting by $\sigma_j$ the standard deviation of feature $j$, we chose $\lambda_j = \frac{1}{2\sigma_j}$ to have a missingness probability that smoothly increases over the support of the data. The bias is then fixed to impose a desired missing rate $r$ based on proposition L.1.

**Proposition L.1** (Achieving a targeted missing rate with probit self-censoring)**.** *Assume that the random variable $X \in \mathbb{R}$ follows a Gaussian distribution and is affected by a probit self-masking mechanism, i.e,*

$$X \sim \mathcal{N}(X|\mu, \sigma^2) \quad and \quad P(M = 1|X) = \Phi(\lambda X - c),$$

*where $\lambda \in \mathbb{R}$ and $c \in \mathbb{R}$ control the slope and shift of the self-masking function. Given a fixed slope $\lambda_0$, a missing rate $r$ is achieved by choosing:*

$$c_0 = \lambda_0 \left( \mu - \Phi^{-1}(r)\sqrt{\lambda_0^{-2} + \sigma^2} \right)$$

*Proof.*

$$r = P(M = 1) \tag{1}$$

$$\iff r = \int P(M = 1|X)P(X)\mathrm{d}X \tag{2}$$

$$\iff r = \int \Phi(\lambda X - c)\mathcal{N}(X|\mu, \sigma^2)\mathrm{d}X \tag{3}$$

$$\iff r = \Phi\left( \frac{\mu - \frac{c}{\lambda}}{\sqrt{\lambda^{-2} + \sigma^2}} \right) \tag{4}$$

$$\iff c = \lambda \left( \mu - \Phi^{-1}(r)\sqrt{\lambda^{-2} + \sigma^2} \right) \tag{5}$$

where eq. (4) is obtained according to equation 4.152 in Bishop. $\qquad\square$

Experiments show that the target missingness probability is achieved even though the features are not Gaussian.

## L.2 RESULTS UNDER MNAR MISSINGNESS.

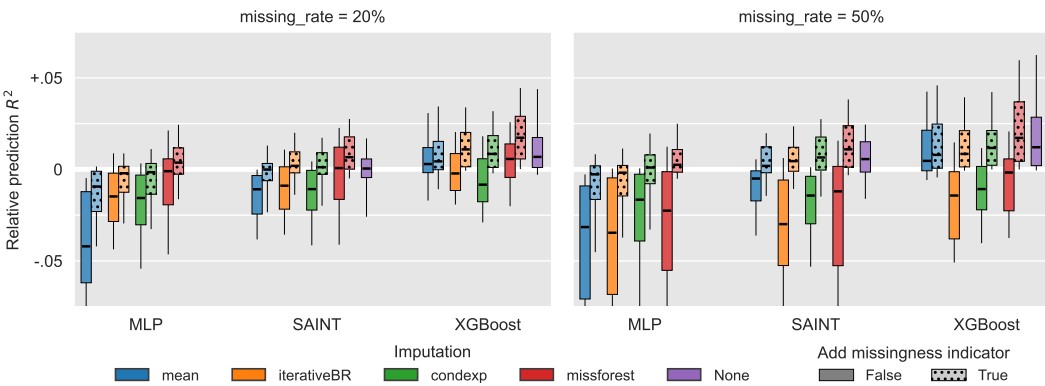

Figure 30: **Relative prediction performances across datasets for different imputations, predictors, and use of the missingness indicator under MNAR missingness.** Each boxplot represents 200 points (20 datasets with 10 repetitions per dataset). The performances shown are $R^2$ scores on the test set relative to the mean performance across all models for a given dataset and repetition. A value of 0.01 indicates that a given method outperforms the average performance on a given dataset by 0.01 on the $R^2$ score. Corresponding critical difference plots in figs. 31 and 32.

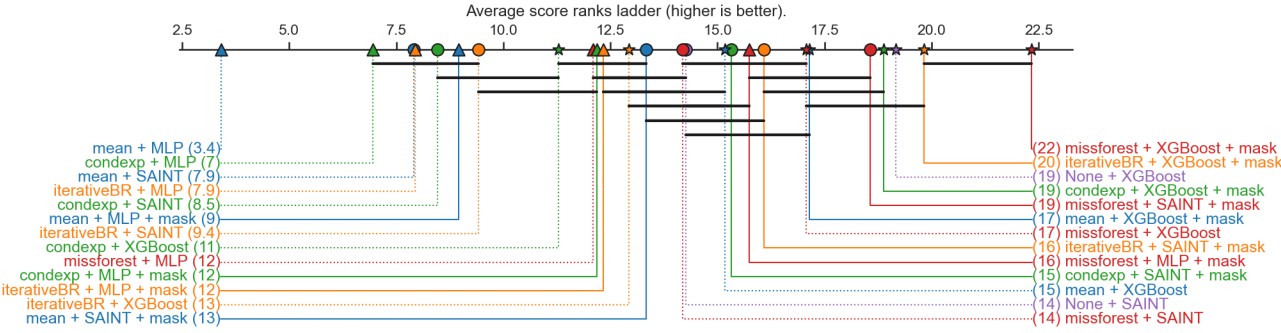

Figure 31: Critical Difference diagram - 20% missingness rate **under MNAR missingness**.

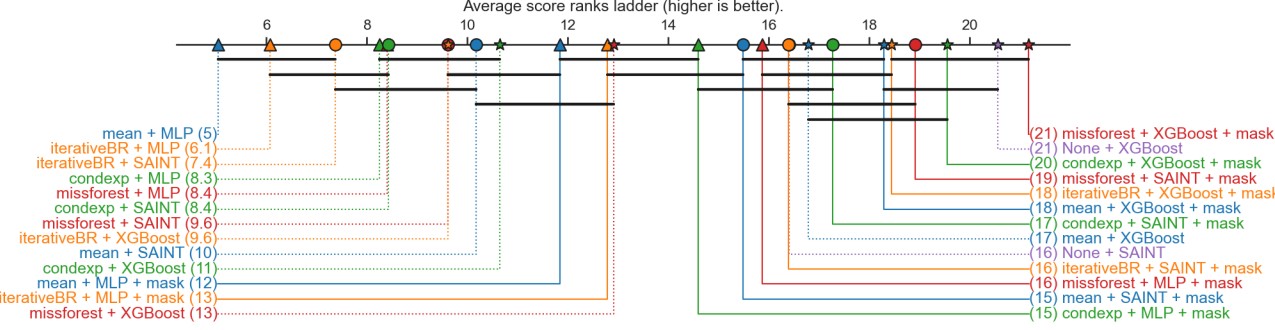

Figure 32: Critical Difference diagram - 50% missingness rate **under MNAR missingness**.

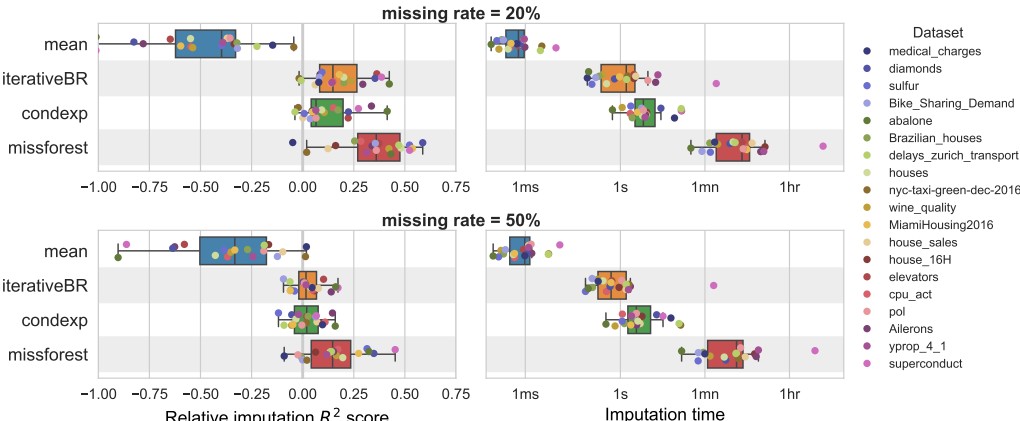

Figure 33: **Left: Imputer performance for recovery under MNAR missingness.** Performances are given as $R^2$ scores for each dataset relative to the mean performance across imputation techniques. A negative value indicates that a method perform worse than the average of other methods. **Right: Imputation time under MNAR missingness.**

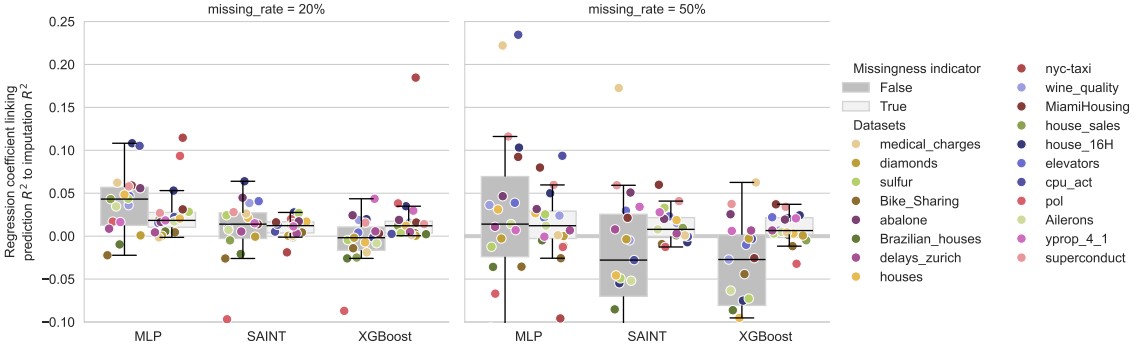

Figure 34: **Effect of the imputation recovery on the prediction performance under MNAR missingness.** We report the slope of the regression line where imputation quality is used to predict prediction performance.

