# OpenReview forum: "Imputation for prediction: beware of diminishing returns."
_ICLR.cc/2025/Conference — ICLR 2025 Spotlight_

### Official Review · Reviewer_NNbq · 2024-10-23

**Soundness:** 4
**Presentation:** 3
**Contribution:** 3
**Rating:** 8
**Confidence:** 4

**Summary:**

This study quantifies the relationship between imputation accuracy and prediction performance across various datasets, showing that improvements in prediction are typically only 10% or less of the gains in imputation \(R^2\). It finds that advanced imputation methods are less beneficial with more flexible models (e.g., XGBoost) and when using missingness indicators, which improve predictions even in MCAR scenarios. The authors highlight that in MNAR settings, the impact of better imputation is likely even smaller.

**Strengths:**

The paper effectively bridges theoretical work on the minimal impact of imputation in asymptotic settings with empirical studies of the "impute-then-predict" pipeline in supervised learning, offering a rigorous evaluation of when imputation accuracy affects prediction. The authors position their work within the existing literature by addressing both empirical evaluations of imputation methods and theoretical frameworks. The evaluation spans 19 datasets, multiple imputation methods, and prediction models across MCAR and MNAR settings, with detailed analysis of relative prediction performance and imputation time. The use of critical difference plots establishes upper bounds on the benefits of imputation, particularly in best-case MCAR scenarios.

**Weaknesses:**

- When drawing conclusions from Figs. 1 and 2 (last paragraph of Sec. 4.1), the authors could address the conditions under which advanced imputers do provide a benefit. For example, while XGBoost might not benefit much from MissForest, other less flexible models like MLP show greater improvements when paired with more advanced imputations.

    The authors might also explore non-linear or hierarchical relationships within the datasets where advanced imputations could outperform simpler methods more consistently. In cases where feature interactions are complex, MissForest could have greater effects.

- Regarding “Good imputations matter less when the response is non-linear” (pgs. 7-8), the authors could strengthen the argument about non-linearities disrupting the relationship between imputation quality and prediction performance by incorporating insights from Le Morvan et al. (2021), which shows that *even with high-quality imputation*, non-linear functions can introduce discontinuities, making them harder to learn optimally. They could further illustrate how the *non-continuous nature of the regression function* after conditional imputation (as discussed in the paper) increases the complexity of the learning process in non-linear settings, which explains why non-linearities amplify the noise in the imputation-prediction relationship.

- On the question of whether the missingness indicator is the optimal way to represent missingness (pg. 9),  the authors could discuss specific limitations of the indicator in more detail, such as potential over-reliance on the indicator or the risk of introducing additional noise into the model. The authors cite the polar encoding paper of Lenz et. al. 2024, but could  suggest additional methods to represent missingness more effectively, such as embedding missingness directly within the model architecture or using neural architectures like NeuMiss that incorporate missing patterns in a more structured way (Le Morvan et al., 2021).

**Questions:**

- Pg. 2., line 70: The MNAR definition could be clarified by stating that missingness is related to the unobserved values themselves, making it informative. In this case, the probability of missing data is related to the actual values that are missing. Also, the MAR and MNAR abbreviations should be spelled out in the first instance.

- Pg. 2, line 103: the sentence could be clarified, e.g., Woźnica & Biecek (2020) trained imputers separately for the training and test datasets, which led to an 'imputation shift'—a situation where the imputation patterns between the two datasets differ, causing inconsistencies in the data used for model training versus model evaluation.

- Pg. 5, it may be informative to state in the “Computational resources” section or in the Appendix the computational hardware (e.g,. number of CPUs) used to conduct the experiments.

**Typos/grammar**:
- Pg. 1, line 52: missing parentheses
- Pg. 2, line 85: “mportant”
- Pg. 5, line 223: It’s been previously stated that for XGBoost and the MLP, hyperparameters are optimized using Optuna (Akiba et al., 2019)
- Pg. 7, line 348: proper casing for figures (“fig. 2”)
- Pg. 8, line 415: citation for Pereira 2024 not included in the reference list

---

> ### Author Response · Authors · 2024-11-26
>
> We thank the reviewer for their time and feedback.
>
> We appreciate the suggested clarifications for various concepts and have incorporated them into our revised manuscript, along with corrections to the identified typos.

---

### Official Review · Reviewer_9esu · 2024-10-30

**Soundness:** 3
**Presentation:** 3
**Contribution:** 2
**Rating:** 6
**Confidence:** 5

**Summary:**

The paper investigates the empirical importance of accurate imputation in tabular, continuous data with MCAR missingness. It compares four common imputation techniques and three model architectures (2x NN, 1x tree-based) across 19 real-world, medium-sized (<50,000 samples), fully-observed datasets augmented with simulated missingness.

The authors conclude that good imputation yields marginal improvements in terms of prediction performance and that missingness indicators may be beneficial even in MCAR scenarios.

**Strengths:**

The study performs an extensive experiment across a variety of imputation methods, prediction models, and datasets.

**Weaknesses:**

The authors conclude that both better imputations and missingness indicators improve prediction performance. Neither of these findings is particularly surprising if we consider the optimal prediction with MCAR data:

$\mathbb{E}[Y \mid X_o, M] = \int_{X_m} f^\star(X_m, X_o)p(X_m \mid X_o)dX_m,$

where, in line with Le Morvan et al. (2021), $Y$ is the outcome, $X_o$ and $X_m$ are the observed respectively missing covariates, and $f^\star$ is the underlying full-data function. For the type of conditional mean imputation used in this work, we do not target $p(X_m \mid X_o)$ but instead estimate $\mathbb{E}[X_m \mid X_o]$. In the extreme, we have perfect knowledge of this expectation through an oracle, and Le Morvan et al. (2021) in their Figure 4 already demonstrated that the performance of such an oracle > imputation via chained equations > mean impute. In the same figure, they also show that missing indicators are beneficial even in MCAR settings. The authors must be aware of this work, since they cite it as their main reference in their section "4.5 WHY IS THE INDICATOR BENEFICIAL, EVEN WITH MCAR DATA?", where they exclusively lean on Le Morvan et al. (2021) to explain why missingness indicators, even if they contain no information about the outcome, may still aid learning an often discontinuous optimal predictor.

References:
Marine Le Morvan, Julie Josse, Erwan Scornet, and Gael Varoquaux. What’s a good imputation to predict with missing values?  Advances in Neural Information Processing Systems, volume 34, pp. 11530–11540. 2021.

**Questions:**

Given that both key findings are consistent with established theory and have been described before empirically, what are the novel contributions that would justify acceptance of the paper?

---

> ### Author Response · Authors · 2024-11-21
>
> We thank the reviewer for their time and feedback.
>
> **TL;DR:** Our keys findings **have not been described empirically before**. The  experiments in [Le Morvan 2021] are on synthetic data and can only serve as illustrations of their theory. They cannot support general practical conclusions, and they do not even reach the same conclusions as ours because of a narrow synthetic focus.
>
> > The authors conclude that both better imputations and missingness indicators improve prediction performance. Neither of these findings is particularly surprising [...]
>
> Our results actually differ from previously published findings, including Le Morvan 2021. Our conclusion is that the impact of imputation on prediction is **very small**, rather than “better imputations improve prediction performances”. The heart of this work is in the estimation of the effect sizes, and observing that in some important cases (powerful models + mask), better imputations do not even improve predictions.
>
> > Le Morvan et al. (2021) in their Figure 4 already demonstrated that the performance of such an oracle > imputation via chained equations > mean impute.
>
> As the reviewer points out, Le Morvan et al. (2021) highlight in their Figure 4 a rather strong positive effect of imputation on prediction, as the prediction performance satisfies: mean imputation < MICE imputation < oracle imputation. This actually contradicts our main message, according to which the effect of imputation accuracy on prediction accuracy is very small. This contradiction is because **their setting is favourable** for imputation to improve prediction:
>
> * The different imputations are combined with a **MLP only** => this is the least powerful model in our study, for which we showed that the potential benefit of imputation is greatest.
>
> * **They work on synthetic data**, where X is Gaussian, and $Y = f(w^\top X) + noise$, with f the square function for example. Although it is not a linear function, it has a specific structure as it is a simple transformation of a linear function, and it is probably closer to our linear setting than to our real data setting. => In our study, linear outcomes yield larger effects than real outcomes.
>
> * Since their data is Gaussian, the imputation task is easier than with real data, leading to large gaps in imputation performance (mean vs oracle).
>
> These points show that their experiments correspond to our most favourable setting for imputation to have an impact on prediction (and even more favourable since they use Gaussian data). We show that in more realistic settings (powerful models + real data), the effect of imputation is greatly minored.
>
> > Given that both key findings are consistent with established theory and have been described before empirically, what are the novel contributions that would justify acceptance of the paper?
>
> Thanks for this important question. The important novel contributions are:
>
> * **Synthetic data illustration versus Extensive real-data benchmark** The main contributions of [Le Morvan 2021] are theoretical results, their empirical results are very limited (only synthetic Gaussian data, with a MLP as prediction model, one missing rate). They cannot support empirical conclusions as our work does.
>
> * **Different empirical conclusions** Because they focus on a synthetic and favourable setting, they report an effect of imputation on prediction that is very optimistic compared to what can be obtained in reality, as we show in our study.
>
> * **An asymptotic theoretical result does not necessarily hold in practice in finite samples** [Le Morvan 2021] conclude  “In further work, it would be useful to theoretically characterize the learning behaviors of Impute-then-Regress methods in finite sample regimes.”, as their main theorem (Theorem 3.1 - Bayes consistency of Impute-then-regress procedures) is asymptotic. This work answers this question with a rigorous empirical study. The fact that our key findings are consistent with established theory is a result, because we did not know what to expect in finite samples.
>
> * **Identification of the factors that modulate the importance of imputation for prediction** Contrary to existing works, we show that more powerful models, appending the mask, and non-linearity of outcomes, all tend to decrease the importance of imputation for better prediction. This is important as focusing on too narrow experimental settings, or averaging over all settings, do not allow to draw actionable and consistent conclusions.
>
> * **Mask in MCAR** Concerning the result that the mask improves prediction even under MCAR data, [Le Morvan 2021] did not report this as a result. The effect of the mask in MCAR with MICE is unclear in their Figure 4, and again the experiments are too limited to allow for empirical conclusions.
>
> Please let us know whether our message and positioning relative to [Le Morvan 2021] makes more sense to you in light of these clarifications. Thanks again for your time.

---

> > ### Comment · Reviewer_9esu · 2024-11-25
> > **Response to the author's comment**
> >
> > I thank the authors for elaborating on their contributions.
> >
> > **Synthetic data illustration versus Extensive real-data benchmark** and **An asymptotic theoretical result does not necessarily hold in practice in finite samples**: Le Morvan's conclude that " almost all imputations lead asymptotically to the optimal prediction, whatever the missingness mechanism". A range of the cited existing benchmarks support this result in the finite samples, showing a limited benefit of imputation and competitive results using simple imputation methods. Papers that come to this conclusion include: Paterakis (2024), Shadbahr (2023), Perez-Lebel (2022), Jäger (2021), and Woznica (2020).
> >
> > **Different empirical conclusions**: I disagree with the authors that Le Morvan 2021 argues for a strong effect of imputation on prediction performance. On the contrary, the best performing model is NeuMiss, a complex model that does not perform imputation, agreeing with the conclusions of this work.
> >
> > **Identification of the factors that modulate the importance of imputation for prediction**:
> > * More powerful models: Shadbahr (2023) investigated the use of more powerful models.
> > * Appending the mask: Paterakis (2024) and Perez-Lebel (2022) mention the importance of appending a missingness mask.
> > * Non-linearity of outcomes: The special case of linear vs. non-linear outcomes has indeed only been discussed theoretically in Le Morvan (2021), but may be of limited interested in real applications.
> >
> > **Mask in MCAR**: Besides Le Morvan (2021), the benefit of missing indicators in MCAR data has also been reported in Paterakis (2024). Although this too was in simulated data, this still demonstrates that a mask can be beneficial in MCAR settings. The general usefulness of missingness indicators in real-world datasets was also shown in Paterakis and Perez-Lebel (2022).
> >
> >
> >
> > In summary, I still do not agree with the authors that their results haven't been described before. Taken in the context of existing works, both theoretical and empirical, their results are not suprising. However, I've come around on the idea that there is sufficient benefit to revisit these questions in a clear and focused way, which the authors have managed in their work. I've changed my score accordingly.

---

### Official Review · Reviewer_RyBn · 2024-10-31

**Soundness:** 3
**Presentation:** 4
**Contribution:** 3
**Rating:** 8
**Confidence:** 4

**Summary:**

This paper explores the link between imputation quality and downstream performance. Through multiple experiments, the paper demonstrates that under MCAR patterns, the quality of reconstruction error is not always linked with performance, depending upon modelling strategies.

**Strengths:**

The paper is well written and easy to read, presenting novel results even under the simple MCAR setting.

**Weaknesses:**

The critical insight that adding missing indicators can be beneficial, even under MCAR, needs more justification (with a potential theoretical justification). Intuitively, I suggest exploring the correlation between reconstruction error and gain from adding the indicator. I believe the correlation should be strong as the model is able to 'discard' badly imputed data. Additionally, an experiment with a random mask appended would demonstrate that this result is not just a product of a larger number of parameters in the model or some regularisation.

Other works have explored this relation. I would recommend comparing the results with work such as Bertsimas 2024, where the authors conclude, "While common sense suggests that a 'good' imputation method produces datasets that are plausible, we show, on the contrary, that, as far as prediction is concerned, crude can be good."

Bertsimas D, Delarue A, Pauphilet J. Simple Imputation Rules for Prediction with Missing Data: Theoretical Guarantees vs. Empirical Performance. Transactions on Machine Learning Research. 2024 Jun 5.

**Questions:**

None

---

> ### Author Response · Authors · 2024-11-25
>
> We thank the reviewer for their time and feedback.
>
> > I suggest exploring the correlation between reconstruction error and gain from adding the indicator.
>
> **Figure 14 in the updated pdf shows the effect of imputation accuracy on the gain from adding the indicator.** We see that most effects are negative, indicating that using the missingness indicator brings the largest boost in prediction performance when imputations have low accuracy. Moreover, effects are strongest for the MLP and smallest for XGBoost, meaning that with more powerful models, prediction boosts due to the missingness indicator are less pronounced. This is coherent with the prediction performances presented in Figure 1. In terms of correlation, the strongest association occurs for the MLP (20% missing rate), with a correlation of -0.5 on average across datasets, and a high dispersion (some datasets have a correlation close to -1, and 3 datasets have a positive correlation).
>
> > Additionally, an experiment with a random mask appended would demonstrate that this result is not just a product of a larger number of parameters in the model or some regularisation.
>
> **We conducted experiments with a shuffled missingness indicator appended (see Figure 15 in the updated pdf)**.
> The columns of the indicator were shuffled for each sample. This preserves the total number of missing values per sample but removes information about which specific features are missing.
> Fig. 15a demonstrates that using a shuffled missingness indicator harms prediction performance, except for XGBoost for which performances are unchanged. In contrast, the true missingness indicator improves performances (fig. 1). Furthermore, the shuffled indicator does not affect the relationship between imputation accuracy and prediction accuracy (fig. 15b), whereas the true indicator reduces the effect size (fig.4).
> These results confirm that the benefit of the missingness indicator is not due to a regularization or merely encoding the number of missing values. Prediction models effectively leverage information about which features are missing, even though under MCAR, this information is unrelated to the unobserved values.
>
> **Relation to Bertsimas (2024)**
> Thank you for highlighting this work. On the theoretical side, Bertsimas (2024) shows that mean imputation is Bayes optimal while mode imputation is not, and clarifies the meaning of "almost all" in the theorem of Le Morvan (2021) regarding the Bayes consistency of Impute-then-Regress procedures. On the experimental side, they compare mean imputation to mode imputation (for categorical features) and MICE (for numerical features) in terms of downstream prediction accuracy. Their findings do not lead to a single conclusion, as the results vary depending on the experimental setting: real vs. (semi-)synthetic data, or linear vs. non-linear outcomes.
> In contrast, we employ multiple imputers to quantify the association between imputation and prediction accuracy, and identify how various factors modulate this relationship (e.g., downstream prediction model, use of the mask, linear vs. non-linear outcomes). Additionally, we explore the effect of using the mask, which was not considered in Bertsimas 2024.

---

### Official Review · Reviewer_JGDK · 2024-11-03

**Soundness:** 3
**Presentation:** 1
**Contribution:** 3
**Rating:** 8
**Confidence:** 4

**Summary:**

The authors use an empirical approach to investigate whether advanced imputation techniques significantly improve predictive accuracy in models with missing data. The study finds that while sophisticated imputation can enhance prediction in certain contexts, the benefits are modest, especially when more expressive models and/or missingness indicators are used. They conclude with the assertion that resources might be better allocated to models that inherently handle missing data rather than focusing extensively on imputation improvements.

**Strengths:**

The value of this paper arises from its significance, not its originality. What I mean by this is that the paper effectively reports a "null" result: something that *doesn't* work. This kind of work is valuable because it saves the community time by not re-inventing square wheels. We need to know what doesn't work.

If you count papers that provide evidence that a method doesn't work (I do) as original, then yes this paper is original. This definition of originality fits under the notion of originality as "a new definition or problem formulation," because the authors are recommending that the data imputation research solves the wrong problem if the goal is improved prediction.

The quality of the work was adequate, they performed a comprehensive study using several different models and many different datasets to arrive at a convincing analysis of model performance.

**Weaknesses:**

I did not mention clarity in the Strengths section because the paper needs a lot of stylistic work. The authors would benefit from reading Strunk & White's "Elements of Style" as well as Steven Pinker's "A Sense of Style." There were many redundant sentences, unnecessary adverbs ("really", "interestingly", etc), unnecessary metadiscourse (e.g., let me tell you what I'm going to tell you) and acronyms that were defined not on their first occurrence (or never at all). Missingness indicators, which are a prominent concept in the paper, should be defined in one sentence in the introduction. In addition, different paragraphs seemed to have different authors, where at least one author appears to not have an adequate grasp of English. I'm not judging that (I don't speak any other language, hats of to them if that is the case) but there are resources to help non-native English speakers (even ChatGPT now does a reasonably decent job: write a paragraph, input it along with "improve:" and then see how the output paragraph is written- I find this helpful for succinctness but it can also help with sentence structure/word choices). All of these weaknesses combined to make reading the paper feel arduous.

**Questions:**

Please eliminate redundant sentences, remove unnecessary adverbs ("really", "interestingly", etc), remove unnecessary metadiscourse (e.g., let me tell you what I'm going to tell you- the end of section 1 ) and acronyms that were defined not on their first occurrence (or never at all). Missingness indicators, which are a prominent concept in the paper, should be defined in one sentence in the introduction. Also, the quality and clarity of the writing varies greatly from paragraph to paragraph - can you make it more consistent?

Line 140: what do you mean by a "universally consistent" algorithm?

Why is the 'semi-synthetic' data (top of page 5) 'semi'? Can you elaborate on that?

Line 341 uses the word "nuanced" in a strange way - so strange I couldn't figure out what the sentence in which it appears actually means.

You say 'comparing imputers is not our main objective' (twice), but it is an important set of results, right? Clearly, it is a salient result that a mean imputer does worse than other methods for both 20% and 50% missingness rates, and you could envision people citing this paper for that result. Perhaps these results should not be downplayed?

Please re-read and consider linearizing the order/structure of your sentences. For example, line 456 reads "For prediction, imputation matters but marginally." You will minimize the amount of cognitive bandwidth / patience of your readers if you linearize the sentence structure, e.g.: 'Imputation matters marginally for prediction." If that is too much of a change of meaning, then consider 'Imputation matters for prediction, but only marginally.' These kinds of reverse-order sentences occur throughout the manuscript and, in aggregate, bog down the reader.

---

> ### Author Response · Authors · 2024-11-26
>
> We thank the reviewer for their time and feedback.
>
> > Line 140: what do you mean by a "universally consistent" algorithm?
>
> It refers to a learning algorithm that achieves asymptotically optimal performance (i.e., minimal possible error) for all data distribution (X, Y) [Gyorfi 2002]. For example, partitioning estimates of the regression function, kernel regression or k-NN regression, are universally consistent.
>
> > Why is the 'semi-synthetic' data (top of page 5) 'semi'? Can you elaborate on that?
>
> We call it semi-synthetic because the response $Y$ is synthetic, but not the design matrix $X$. Specifically, $X$ is derived from real datasets in our benchmarks, while the response variable $Y$ is simulated as a linear function of $X$.
>
> > Line 341 uses the word "nuanced" in a strange way - so strange I couldn't figure out what the sentence in which it appears actually means.
>
> We changed the original sentence:
>
> *However, this observation should be nuanced by the size of the effects.*
>
> to this one:
>
> *However, this observation should be interpreted with nuance in light of the effect sizes.*
>
> We hope this clarifies the intended meaning: effect sizes should be considered to avoid overemphasizing the positive impact of imputation on prediction.
>
> **About imputer comparison**
>
> > Perhaps these results should not be downplayed?
>
> Thank you for this comment. We initially chose not to emphasize these results, as they are not central to our main message. It could also be argued that the mean imputers' poorer performance is expected. Based on the reviewer’s feedback, we removed one of the sentences that downplayed this result.
>
> **About the writing**
>
> We thank the reviewer for their advice. We fixed the acronym definitions, added the definition of the missingness indicator, removed some redundancies, removed some unnecessary words such as ‘really’ and ‘interestingly, and went through the manuscript to try to linearize sentences.
>
> [Gyorfi 2002] A distribution-free theory of nonparametric regression.

---

### Official Review · Reviewer_ETS8 · 2024-11-10

**Soundness:** 2
**Presentation:** 3
**Contribution:** 2
**Rating:** 6
**Confidence:** 3

**Summary:**

This paper empirically studies the effects of various missing value imputation methods, in particular whether better imputation accuracy yields higher prediction performance. The authors conduct experiments on nineteen benchmark datasets using four imputation methods, three different models, as well as variations such as missingness indicators and semi-synthetic linear labels. They conclude that imputation accuracy does affect prediction performance, although the gains are quite small, and the effect is further reduced if the model is more expressive, missing value indicators are used, or the target variable has a non-linear relationship to the covariates.

**Strengths:**

The paper presents an empirical study that can help us better understand the effect of various imputation techniques for different model/problem scenarios, which is an important question as missing values are ubiquitous and often handled by imputation. The paper specifically considers the link between imputation accuracy and prediction performance, which have been studied theoretically in some scenarios but not empirically for real-world problems.

The experimental setup is clearly described in detail for reproducibility, and the authors mention that code will also be available upon publication. The experiments are also thorough in terms of datasets, models, and imputation methods.

Related work discussion is clear, including both theoretical and empirical studies about missing value imputation methods.

**Weaknesses:**

Many claims/conclusions in the paper are based on small differences in average values with overlapping confidence intervals, and there are no statistical tests for significance. For example, the authors claim that the imputation techniques considered in the experiments show diverse imputation performance (Figure 2), but apart from mean imputation being worse than others, there doesn’t seem to be a definitive difference in quality by the other three imputers.

In addition, the claim about correlation of imputation accuracy and prediction performance is made based on the small but positive slope of the regression line (Figure 4). However, the slope seems very small in most cases to suggest this, especially in the 20% missing case, and the correlation may still be very weak (small correlation coefficient). On the other hand, the authors suggest that good imputations matter more in the linear response case (semi-simulated data), based on the slightly higher correlation (Fig 5), but the slope seems still very small (Fig 11).

In about a third of the benchmark datasets, the dimension is small such that 20% missing means just one missing feature. I would expect the difference between simple and more complex imputation techniques to be pretty small in such cases.

The experiment in MNAR scenario (Section 4.4) didn’t feel very connected to the rest of the paper. As the authors also noted, it is well known that most imputation methods (without considering causal structures) are not valid under MNAR.

-------------
Post-rebuttal: Thanks to the authors for the detailed response and revision. I think the additional experiments and statistical tests made the results of this paper much more convincing, and I am happy to raise my score to 6.

**Questions:**

The observation about good imputations having less effect when using missingness indicators was very interesting. To test the intuition mentioned in Section 4.5, I think it would be interesting to use explainability techniques on these models trained with missingness indicators to see if the importance/weights of features drop when they are missing (i.e. inputs with the corresponding missingness indicator on).

Does the correlation of prediction performance and imputation accuracy (e.g, Figures 3 and 4) also show any difference by imputation method?

Would larger missingness rates show more significant correlations?

---

> ### Author Response · Authors · 2024-11-21
>
> We thank the reviewer for their time and feedback.
>
> **Regarding the weaknesses reported**
>
> The reviewer is concerned by the small effect sizes reported, and questions the statistical significance of some conclusions. We acknowledge that the reported effect sizes of imputation on prediction accuracy are small—this is indeed one of our key messages. The reviewer rightly highlights the need for statistical support for finer-grained claims (e.g., effects are small but non-zero). We confirm that our conclusions are supported by statistically significant differences (details of the tests provided in a separate comment):
>
> * **Small but non-zero effect sizes**: Effects are significantly greater than 0 across datasets for less powerful models (e.g. MLP) but not for more powerful ones (e.g. XGBoost). Looking at individual datasets reveals heterogeneity: some datasets show statistically significant effects, while others do not. In an updated Figure 4 (see updated pdf), round shapes indicate coefficients significantly greater than 0 (one-sided T-tests, pval < 0.05, Bonferroni correction), while triangles denote non-significant coefficients. Overall, in the best case (MLP, no mask, 20% missing rate), the effect of imputation on prediction is significantly > 0 for 13/19 datasets. In the worst case (XGBoost, no mask, 50% missing rate), this holds for 6/19 datasets.
>
> * **Higher correlation with linear outcomes**: Wilcoxon signed-rank test pval=1e-19. This translates into more coefficients significantly > 0 in the linear case than in the real outcome case.
>
> * **Larger effects without the mask**: Wilcoxon signed-rank test pval=1e-10.
>
> These findings are consistent across experiments and statistically significant, reinforcing our overall message about the diminishing returns of imputation.
>
> **Regarding the missing rates**
>
> > In about a third of the benchmark datasets, the dimension is small such that 20% missing means just one missing feature.
>
> Each entry has a 20% chance of being missing, rather than each sample has 20% of its entries missing, we will clarify this in the paper. Therefore the number of missing features in a sample varies according to a binomial distribution. This creates more diversity than forcing a 20% of missing values per sample.
>
> > Would larger missingness rates show more significant correlations?
>
> We ran supplementary experiments at a 70% missing rate on all datasets (on top of the 20% and 50% missing rates), and added **a new figure in appendix which plots the effects and correlations as a function of the missing rates** (Figure 12 in the updated pdf) for the various models. We see that effects are larger for higher missing rates: this is particularly clear for linear outcomes, but less for real outcomes. This suggests that imputation matters more at higher missing rates, although for real outcomes and powerful models, these effects are still very small. By contrast, correlations decrease when the missing rate increases, i.e, the association is noisier (less likely to be significant).
>
> **Regarding feature importances**
>
> > “use explainability techniques on these models trained with missingness indicators to see if the importance/weights of features drop when they are missing”
>
> We computed feature importances using feature permutation (scikit-learn's permutation_importance). For each dataset, we identified the two most important features on the test set. For each feature, we then re-calculated its importance across the subset of test samples for which the feature was missing, and across the subset where it was observed. This allows comparing the importance of a feature when it is missing versus when it is observed. This experiment was conducted using XGBoost with condexp or missforest imputation, both with and without the mask.
>
> **Results are shown on a new Figure (Figure 13 in the updated pdf)**, and indicate that on average, a feature is half as important when imputed compared to when observed, with considerable variability (i.e., many features are 10 times less important when imputed, and some features remain as important when imputed). When a mask is used, importances drop significantly more with missforest imputation compared to when no mask is used (Wilcoxon signed-rank test p-value < 0.01). However, this effect is not observed with condexp imputation.
>
> > Does the correlation of prediction performance and imputation accuracy (e.g, Figures 3 and 4) also show any difference by imputation method?
>
> A given imputation method gives very similar accuracies when repeated on the same dataset with a new sampling of missing values, therefore looking at the correlation between imputation and prediction per imputation methods is not informative: there is too little variability in the imputation accuracy.

---

> > ### Author Response · Authors · 2024-11-21
> > **Details about the statistical tests**
> >
> > **Statistical significance of the small positive effects of imputation on prediction (Fig.4)**
> >
> > We investigated the p-values associated with each coefficient, testing the hypothesis (T-test): is the estimated coefficient > 0? For all tests, we used a significance level of 0.05 with a Bonferroni correction for multiple testing, thus obtaining conservative conclusions.
> >
> > To summarize these results, we did a one-sided Wilcoxon signed-rank test, testing whether the median of each boxplot in Fig.4 is significantly greater than 0 (significance level of 0.05 with Bonferroni correction). At 20% (resp 50%) missing rate, the impact of imputation on prediction is NOT significantly greater than 0 across datasets for SAINT and XGBoost (resp. SAINT, XGBOOST, XGBoost + mask) but significant for all other models and mask combinations.
> >
> > **Statistical significance of the real versus linear outcome case**
> >
> > >The authors suggest that good imputations matter more in the linear response case (semi-simulated data), based on the slightly higher correlation (Fig 5), but the slope seems still very small (Fig 11).
> >
> > A Wilcoxon signed-rank test (pval=1e-19) shows that correlations are significantly larger for linear outcomes than for matched real outcomes. This means that there is significantly less noise in the relationship between imputation and prediction accuracies, which translates into more coefficients that are significantly greater than 0 in the linear case than in the real outcome case (see Fig. 4 vs Fig 11 in the updated pdf). This is why we say that “Good imputations matter less when the response is non-linear.”
> >
> >
> > **Statistical significance of imputation quality differences (Fig. 2)**
> >
> > * p-values for the Wilcoxon signed-rank test - 20% missing rate:
> >
> > |              | condexp | iterativeBR | mean      | missforest |
> > |--------------|---------|-------------|-----------|------------|
> > | condexp      |         | 0.418041    | 0.000004  | 0.000072   |
> > | iterativeBR  |         |             | 0.000004  | 0.000336   |
> > | mean         |         |             |           | 0.000004   |
> > | missforest   |         |             |           |            |
> >
> > Only the difference between condexp and iterativeBR is not statistically significant (these two were described as tied in the paper).
> >
> > * p-values for the Wilcoxon signed-rank test - 50% missing rate:
> >
> > |              | condexp | iterativeBR | mean      | missforest |
> > |--------------|---------|-------------|-----------|------------|
> > | condexp      |         | 0.000004    | 0.000004  | 0.054573   |
> > | iterativeBR  |         |             | 0.000038  | 0.312408   |
> > | mean         |         |             |           | 0.000126   |
> > | missforest   |         |             |           |            |
> >
> > iterativeBR is not statistically different from missforest, which was also expected from a visual inspection. The hypothesis that condexp and missforest are on par cannot be rejected either. All other pairs have significantly different performances.

---

> > ### Author Response · Authors · 2024-11-29
> >
> > In our rebuttal, we have provided new experiments addressing the reviewer’s questions:
> >
> > * On the feature importances of imputed vs. observed features.
> > * On the effect of the missing rate.
> > * On the significance of the observed effects.
> >
> > We would be grateful if the reviewer could share their updated opinion of our work based on this additional evidence.
> >
> > Thank you again for your time and consideration.

---

### Meta-Review · Area_Chair_Sku4 · 2024-12-21

**Metareview:**

The paper proposes an empirical study that analyses the effectiveness of advanced imputation methods for missing values.
All reviewers are happy about the findings of the paper and agree to accept the paper.

**Additional Comments On Reviewer Discussion:**

Reviewers were happy about the rebuttals and some of them raised their evaluation of the paper

---

### Decision · Program_Chairs · 2025-01-22

Accept (Spotlight)